# An approach for constraining mantle viscosities through assimilation of paleo sea level data into a glacial isostatic adjustment model

Reyko Schachtschneider[1], Jan Saynisch-Wagner[1], Volker Klemann[1], Meike Bagge[1], and Maik Thomas[1,2]

[1]Helmholtz Centre Potsdam GFZ German Research Centre for Geosciences, Telegrafenberg, 14473 Potsdam, Germany
[2]Freie Universität Berlin, Kaiserswerther Str. 16-18, 14195 Berlin, Germany

**Correspondence:** Reyko Schachtschneider (reyko.schachtschneider@gfz-potsdam.de)

**Abstract.**

Glacial isostatic adjustment is largely governed by rheological properties of the Earth's mantle. Large mass redistributions in the ocean-cryosphere system and the subsequent response of the viscoelastic Earth have led to dramatic sea level changes in the past. This process is ongoing and in order to understand and predict current and future sea level changes the knowledge of mantle properties such as viscosity is essential. In this study we present a method to obtain estimates of mantle viscosities by assimilation of relative sea level rates of change into a viscoelastic model of the lithosphere and mantle. We set up a particle filter with probabilistic resampling. In an identical twin experiment we show that mantle viscosities can be recovered in a glacial isostatic adjustment model of a simple three layer earth structure consisting of an elastic lithosphere and two mantle layers of different viscosity. We investigate the ensemble behaviour on different parameters in three setups: (1) global observations dataset since last glacial maximum with different ensemble initialisations and observation uncertainties, (2) regional observations from Fennoscandia or Laurentide/Greenland only, (3) limiting the observation period to 10 ka BP until present. We show that the recovery is successful in all cases if the target parameter values are properly sampled by the initial ensemble probability distribution. This even includes cases in which the target viscosity values are located far in the tail of the initial ensemble probability distribution. Experiments show that the method is successful if enough near-field observations are available. This makes it work best for a period after substantial deglaciation until present when the number of sea level indicators is relatively high.

## 1  Introduction

Glacial isostatic adjustment (GIA) describes the continual response of the Earth to mass redistribution between continental glaciers, ice sheets and the ocean during glacial cycles (e.g., Lambeck et al., 2003). These quasi-periodic mass redistributions occur due to climate cycles that have their origin in astronomical cycles of precession, obliquity, and eccentricity with periods near 23,000, 41,000, and 96,000 years (Imbrie et al., 1992). In the past, deformation of the Earth's surface due to those mass redistributions have led to raising and falling sea levels with local amplitudes exceeding a hundred meters (e.g. Whitehouse,

2018; Lambeck et al., 2014; Haskell, 1935). The behaviour of sea level during glaciation and deglaciation is very complex and differs in the near-, intermediate-, and far-field (Khan et al., 2015).

Understanding GIA processes is essential for the quantification of past and recent sea level changes. Especially, the rheology of the Earth's mantle plays a significant role in surface deformation in the near-field of ice sheets (Lambeck et al., 1998). Therefore, the Earth's response after deglaciation is one important process that allows to infer mantle viscosities (Steffen and Wu, 2011; Peltier, 1996). Obtaining reliable values for mantle viscosity is the basis for a precise determination of Earth's deformation history, mass re-distribution, and sea level changes. Especially, when estimating the distribution of local sea level

change due to ongoing melting of glaciers and continental ice sheets in Greenland and Antarctica, the precise knowledge of isostatic adjustment processes is indispensable. In the far-field mantle rheology plays a minor role. Here, the barystatic-GRD fingerprints (GRD refers to changes in Earth Gravity, Earth Rotation and viscoelastic solid-Earth Deformation), i.e. the sum of elastic deformation, change of the geoid, and barystatic sea level rise (Gregory et al., 2019), are dominant.

There have been a large number of studies that use different techniques and data to infer the viscosity structure of the Earth's

mantle. Peltier (1974) determined the impulse response of a viscoelastic Earth to changing mass load and applied the findings to GIA (Peltier and Andrews, 1976). Later, the solution of the sea level equation (SLE) was included in the calculations of surface deformation (Peltier et al., 1978; Clark et al., 1978). That approach was further improved by, e.g. Mitrovica et al. (1994) who developed spectral methods solving the SLE and also combined complementary data such as mantle convection with GIA to obtain mantle viscosity profiles (Mitrovica and Forte, 2004, 1997). Lambeck (1993) developed a global GIA model that

accounts for ice and ocean loading as well as shoreline migration. Bagge et al. (2021) used geodynamically constrained global viscosity structures to investigate the effect of lateral variations on sea level predictions. Steffen and Kaufmann (2005) applied an algorithm based on the neighbourhood algorithm to use paleo shoreline data and radial crustal velocities obtained from GPS measurements for inferring the viscosity beneath Scandinavia and northwestern Europe. Another computationally efficient approach using sensitivity kernels (sensitivity of observations to both mantle viscosity and ice load history) was presented

by Al-Attar and Tromp (2014) who applied an adjoint method to the viscoelastic loading problem. Caron et al. (2017) used a Bayesian Monte Carlo approach and >5000 global paleo sea level data to evaluate the range and probability of possible solutions for different ice histories and Earth structures. A comprehensive overview over the development of GIA modeling can be found in e.g. Whitehouse (2018), Steffen and Wu (2011), and Whitehouse (2009). Other data assimilation techniques have been applied to deformation models (Hill et al., 2010) and the estimation of the sources of sea level rise (Hay et al., 2013).

In this study, we present a method that allows to draw conclusions about mantle viscosity values by assimilating relative sea level (RSL) observations into a viscoelastic Earth model. In an identical twin experiment we assimilate RSL rates computed from a reference model. We apply a particle filter and study how the mean parameter state of the model ensemble converges to the target parameter state. First, we demonstrate the applicability of the method and then show two special cases that are relevant for assimilation of real RSL observations.

The paper is organised as follows. In section 2 the viscoelastic deformation model is described. Section 3 deals with the basic principles of data assimilation and the particular filter that was used in this study. The experiment setup description follows in

**Table 1.** Depth structure of VILMA model in this study with number of spectral-finite elements (SFEs) in vertical direction per region and viscosity of the reference model.

| Region | Depth [km] | Number of SFEs | Viscosity [Pa s] |
|---|---|---|---|
| Lithosphere | 0 – 60 | 12 | $10^{30}$ |
| Upper mantle | 60 – 670 | 97 | $10^{20}$ |
| Lower mantle | 670 – 3,891 | 55 | $10^{21}$ |

Sect. 4. Our results are presented and discussed in Sect. 5 and Sect. 6, respectively, followed by some concluding remarks and an outlook in Sect. 7.

## 2 Glacial Isostatic Adjustment Model

In this study, we use the VIscoelastic model of the Lithosphere and MAntle (VILMA) (Klemann et al., 2008). In VILMA, the continuum mechanical equations of a self-gravitating, viscoelastic and incompressible Earth are solved in the time domain following the spectral–finite element (SFE) formulation of Martinec (2000). See appendix A for the basic equations of this method. The model is capable of handling 3D viscosity distributions (Klemann et al., 2008). The sea level change is determined with the sea level equation of Farrell and Clark (1976) (cf. appendix B), i.e. considering a mass conserving and gravitationally

consistent redistribution of water, on a deformable ground. In particular, for the water-mass redistribution between ice-sheets and ocean, the effects of moving coast lines and floating ice are considered (Hagedoorn et al., 2007), which is consistent with (Kendall et al., 2005). The main difference to standard Laplace-domain formulations based on Peltier (1974) is the solution in the time domain, which allows a direct update of the viscoelastic model state during the assimilation process. This is described in more detail in Sect. 3.

We consider a 1D Earth structure, i.e., the mantle viscosity varies only with depth, and is set to respective constant values of $10^{20}$ and $10^{21}$ Pa s for the upper and the lower mantle, whereas the lithosphere is considered as an elastic layer of 60 km (cf. Table 1). The given values were considered ad hoc and not as values representing a realistic viscosity structure in order to emphasise the synthetic character of this experiment. The fluid core is considered as a lower boundary condition. Shear modulus and density follow the elastic PREM structure (Dziewonski and Anderson, 1981). The thickness of the radial finite

elements is ranging between 40 km at the base of the lower mantle to 5 km in the lithosphere and sum up in total to 164 SFEs in the vertical (cf. Table 1). In horizontal directions the problem is solved in spherical harmonics, degree and order ranging from 0 to 170. The load is represented as a $256 \times 512$ Gauss-Legendre grid, on which the sea level equation is solved.

As forcing, the surface mass load of the last glacial cycle in the parameterization of the ICE-5G reconstruction (Peltier, 2004) is considered, which covers the time range from 123 ka BP (before present) to present day. The integration time step

was set to 20 years.

## 3   Data Assimilation

### 3.1   General

Data assimilation provides a way to combine dynamic models with observations (Asch et al., 2016). Using data assimilation techniques, a model can be updated based on observations in order to obtain new model parameters that better explain the data. There are various data assimilation techniques that are appropriate for certain applications or scenarios. They have been used in a wide range of scientific fields, including numerical weather prediction (Bauer et al., 2015), ocean circulation modelling (Saynisch et al., 2015; Irrgang et al., 2017), geomagnetic field modelling (Bärenzung et al., 2018) and geodynamo studies (Fournier et al., 2013).

A well-known method for solving non-linear filtering problems with non-Gaussian error statistics is the extended Kalman filter by Anderson and Moore (1979). Its principle is based on linearisation of evolution models with Taylor series expansions. Such approximation can lead to poor representations of the models non-linearities and its probability density function (PDF) and the filter can diverge (Van der Merwe et al., 2001).

In our study, we used the particle filter (see Sect. 3.2) for parameter estimation in the dynamic model VILMA. The goal of parameter estimation is to find a set of parameters in a model that leads to a solution that is consistent with a set of observations (Evensen, 2009). The parameters we attempt to estimate are the viscosities of the lower and upper mantle.

### 3.2   The particle filter

The particle filter is an ensemble-based data assimilation technique. It follows the Monte-Carlo view that any probability distribution can be represented by a discrete sample from that distribution (Liu et al., 2001). Ensemble based techniques are particularly useful if the models are non-linear and the PDF of the model state or the errors are not Gaussian (Van Leeuwen, 2009). It allows a complete representation of the posterior model state distribution so that statistical estimates are easy to compute (Van der Merwe et al., 2001). In a particle filter, each ensemble member is assigned a weight factor which is updated during each assimilation step. The output of the filter is the weighted posterior ensemble PDF. From that PDF a weighted mean can be calculated. If the posterior ensemble PDF is strongly non-Gaussian, the weighted mean can be a poor estimate. Due to the nature of our particle perturbation this is not the case in our study. In particle filters without resampling, after some assimilation steps the ensemble can contain particles with very low weights that are practically insignificant (Pham, 2001). Therefore, without resampling large ensembles are needed in order to properly sample the model PDF. If too few particles with significant weight remain, the filter is degenerated and does not represent the model PDF. Therefore, if particles with low significance are resampled to particles with higher significance, the ensemble size can be reduced while the ensemble PDF can still represent the model PDF. More profound introductions to data assimilation and particle filters can be found, e.g., in Asch et al. (2016), Carrassi et al. (2018), Fearnhead and Künsch (2018), and Van Leeuwen et al. (2019).

The model update in our particle filter is based directly on Bayes' theorem (for an introduction see e.g. (Box and Tiao, 2011). If the PDFs of the model and the PDFs of the observations are continuous, Bayes' theorem holds:

$$p_{\mathrm{m}}(\boldsymbol{\psi}|\boldsymbol{d}) = \frac{p_{\mathrm{d}}(\boldsymbol{d}|\boldsymbol{\psi})p_{\mathrm{m}}(\boldsymbol{\psi})}{p_{\mathrm{d}}(\boldsymbol{d})} \; , \tag{1}$$

where $p_{\mathrm{m}}(\boldsymbol{\psi}|\boldsymbol{d})$ is the posterior PDF of the model given the observations, $p_{\mathrm{d}}(\boldsymbol{d}|\boldsymbol{\psi})$ is the likelihood of the observations given the model, $p_{\mathrm{m}}(\boldsymbol{\psi})$ is the prior PDF of the model, and $p_{\mathrm{d}}(\boldsymbol{d})$ a normalisation factor, the so-called model evidence. The weights of the $i$th particle are given by

$$w_i = \frac{p_d(\boldsymbol{d}|\boldsymbol{\psi}_i)}{\sum_{j=1}^{M} p_d(\boldsymbol{d}|\boldsymbol{\psi}_j)} \; , \tag{2}$$

where $M$ is the ensemble size and the likelihood is given by

$$p_{\mathrm{d}}(\boldsymbol{d}|\boldsymbol{\psi}) = \exp(-0.5\,\boldsymbol{r}^T \mathbf{R}^{-1} \boldsymbol{r})$$

with $\boldsymbol{r}$ the vector of observation residuals, i.e. the differences between observations and model predictions, and $\mathbf{R}$ the observation error covariance matrix. The weights $w_i$ represent the significance of a particle, i.e. its contribution to the estimate of the ensemble mean, and determine the chance for survival in the resampling step.

We use a particle filter with importance resampling and perturbation. Its principle is illustrated in Fig. 1. Sequential Importance Resampling (SIR) was proposed by Rubin (1988) and applied to filtering of dynamical systems by Gordon et al. (1993). In this approach, an ensemble of model realisations is propagated in time. When observations become available, each ensemble member's performance is evaluated based on the differences between the observations and the corresponding values computed from the model states. These measures are used to decide which particles further propagate and which are disregarded after that assimilation step. Particles with low performance are resampled to states of better performing particles. Thus, the ensemble size stays constant throughout the entire assimilation run.

Weighting the particles reduces the ensemble variance. If the variance becomes very small there is the risk of filter degeneracy (Fearnhead and Künsch, 2018). A particle filter has degenerated if almost all ensemble members have very low or zero weight and the model PDF is no longer represented by the ensemble. In that case, when resampling the particles, all particles with very low weights would be resampled to (the very few) states of particles with large weights. The main approaches to overcome degeneracy are (Fearnhead and Künsch, 2018): firstly, adding a random value to the estimated parameters (e.g. Liu and West, 2001); secondly, using a Monte-Carlo Markov Chain within the particle filter (Fearnhead, 2002) in order to obtain new parameter values; and thirdly, using a stochastic approximation method in which the particle filter update depends on the current parameter estimate (e.g. Poyiadjis et al., 2011). Another approach is jittering. It involves building a suitable kernel estimator of the posterior PDF given the observations from which new samples of the parameter space can be drawn (Liu and West, 2001). However, jittering has been successfully applied only to cases with low-dimensional state space (Crisan and Miguez, 2018). Of the mentioned methods, only the second approach gives the exact solution while the first and third are approximations. We follow the first approach (addition of random perturbations) since it is simple to implement and allows us to enhance the filter convergence by constraining the perturbation. While Liu and West (2001) proposed shrinking the ensemble

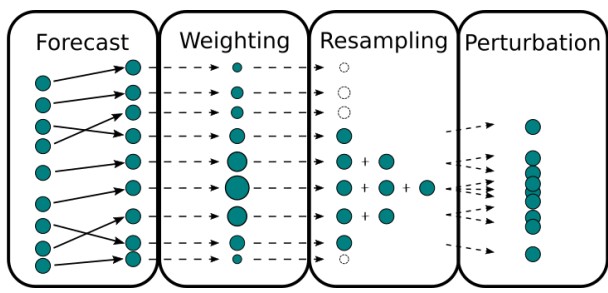

**Figure 1.** The particle filter principle. In the forecast phase the dynamic model ensemble is propagated in time until observations become available. Then, the ensemble members are assigned a weight factor based on the residuals between the observations and the according values computed from the model states. Based on the weight the ensemble is resampled. Members with low weights are disregarded, members with high weights are copied. The ensemble size stays constant. Finally, the model states of the ensemble members are perturbed and the next model integration cycle starts with the updated ensemble.

to its mean state before adding noise, we add noise directly to the resampled particles. After resampling the particles, a random value based on the current ensemble variance is added to each particle's mantle viscosity values. The random values are drawn

from scaled normal distributions $N(0, a^2 \sigma_{\mathrm{U,L}}^2)$ for the two mantle regions separately, where $\sigma_{\mathrm{L}}^2$ and $\sigma_{\mathrm{U}}^2$ are the ensemble variances for the lower and upper mantle, respectively. In the case that the resulting perturbed viscosity was negative, the absolute value was chosen. The scaling factor $a$ is introduced to control the convergence of the ensemble. It is set to $0.5$ in this study. This increases the variance by 25% after weighting and resampling.

Figure 2 illustrates probabilistic resampling as used in our filter. After the forecast, the particle weights (cf. Eq. (2)) estimate

how probable a particle is given the observations (Carrassi et al., 2018). For resampling, a value $w_i' = \sum_{j=1}^{i} w_j$ is assigned to each particle. Then, $M$ random numbers $r_i$ are drawn from a uniform density on [0,1]. The $i$-th particle is resampled to the model state of the $j$-th particle if $w_{j-1}' < r_i \leq w_j'$. Stochastic Universal Resampling (Kitagawa, 1996) would be a possibly faster method since the draw of only one random number is required but it would allow also particles with low weight to survive. Due to the addition of quite large perturbations after resampling the differences are negligible.

For implementing the particle filter into the VILMA model the Parallel Data Assimilation Framework (PDAF) by Nerger et al. (2005) is used. It is a versatile software package that helps to include a variety of data assimilation techniques into pre-existing model codes.

## 4 Experiment Setup

### 4.1 General

The experiments we present are conducted as sandbox experiments. We use an identical twin setup for the reference model run and the assimilation simulation. Each ensemble is initialised from the model state of the reference run $m_0$ at time $t_0$. This is

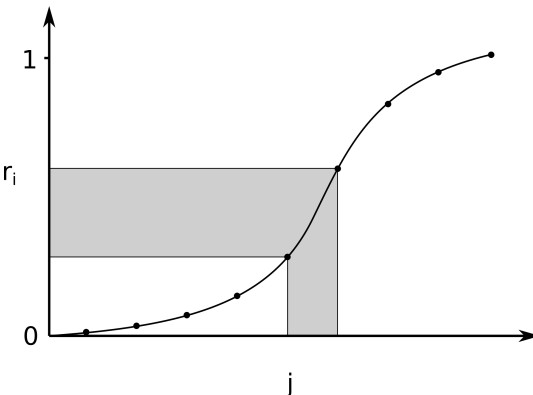

**Figure 2.** Resampling principle. After drawing a random number $r_i$ from a uniform distribution on [0,1] the $i$-th particle is resampled to the model state of the $j$-th particle if $r_i$ falls in the corresponding bin in the cumulative distribution of the normalised weights.

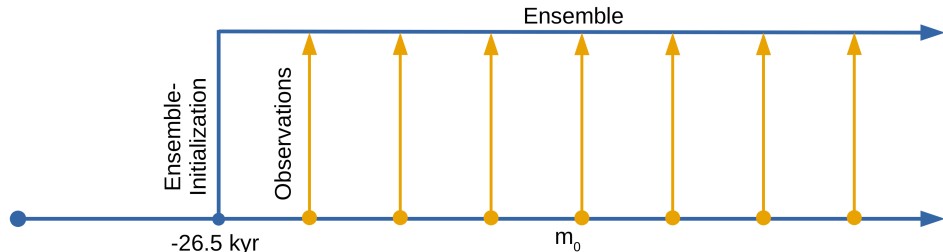

**Figure 3.** Setup of the identical twin experiment. The ensemble of particles is initialised from the model state of the target run $m_0$ at $t_0 = -26.5$ ka. During the assimilation steps, observations of RSL rates, obtained from the target run, are assimilated into the ensemble. This is done every 1 kyr.

done by adding random values drawn from a normal distribution $N(\mu_{\mathrm{init}}, \sigma^2_{\mathrm{init}})$ to the viscosity values $\nu$ of the lower and upper mantle, respectively, for each ensemble member. The mean $\mu_{\mathrm{init}}$ is called initial offset. The reference run's mantle viscosity values function as target values for the assimilation experiments (cf. Table 1). All other model parameters remain unchanged

during the assimilation. For the values governing the ensemble initialisation, see Table 2. The RSL values determined by the reference model at respective locations and times are used to calculate the RSL rates that constitute the synthetic observations used in the assimilation.

    In Fig. 3, the sandbox experiment principle is illustrated. From the initialisation at $t_0$, the ensemble is propagated in time. At times $t_n = t_0 + n\Delta t$, when synthetic observations are available, they are assimilated and the model ensemble is updated

(cf. Sect. 3.2). The interval between consecutive observations is $\Delta t = 1\,\mathrm{kyr}$. Observations are continuously assimilated into the ensemble and the convergence of its weighted mean viscosity values (cf. Eq. (2)) to the values of the reference run, $m_0$, are investigated.

**Table 2.** Parameters of the test cases investigated, with standard deviation of RSL observations ($\sigma_{\mathrm{obs}}$), mean and standard deviation of ensemble perturbation at initialisation ($\mu_{\mathrm{init}}$ and $\sigma_{\mathrm{init}}$), regions from which observations were used and the time interval of observations. Pairs of values represent lower / upper mantle viscosities. The observation uncertainty of case E is variable and decreases with time from 7 m/ka to 0.5 m/ka, details in the text.

| Parameter / Case | A | B | C | D | E | LG | FS | B10 | C10 |
|---|---|---|---|---|---|---|---|---|---|
| $\sigma_{\mathrm{obs}}$ [m/ka] | 0.1 | 0.25 | 0.5 | 0.25 | var. | | 0.25 | | 0.5 |
| $\mu_{\mathrm{init}}$ [Pa s] | | 2.0e20 /2.0e19 | | 4.0e20 / 4.0e19 | | | 2.0e20 / 2.0e19 | | |
| $\sigma_{\mathrm{init}}$ [Pa s] | | | | | 2.0e20 / 2.0e19 | | | | |
| Regions | | | global | | | Laurentide+ Greenland | Fenno- scandia | global | |
| Obs. period [ka BP] | | | | | 25.5 – 0.5 | | | 9.5 – 0.5 | |

As a starting point for the assimilation we chose $t_0 = 26.5$ ka BP and 10.5 ka BP, respectively. The date 26.5 ka BP marks the beginning of the last glacial maximum (LGM). The date 10.5 ka BP approximately marks the end of the last deglaciation. Thereafter, the number of available observations increases significantly. Those setups are meaningful since the first setup gives large signals in RSL change while in the second setup more observations are available.

## 4.2 Synthetic observations

Geological sea level index points (SLIPs) allow to reconstruct the relative sea level (RSL) during the glacial cycle. RSL is defined as the change of the water height, $S$, measured relative to the earth surface or ground, $T$, and referenced to the present state, $t = 0$ (see appendix B for a more comprehensive derivation of the sea level equation):

$$h_{\mathrm{rsl}}(t) = S(t) - T(t) - [S(t=0) - T(t=0)]$$

However, when modelling RSL in the forward sense considered here, $S(t) - T(t)$ is specified relative to the initial state. The referencing to $S(t=0) - T(t=0)$ can only be performed at the end of the integration, and, in consequence, its actual value will depend on the considered parameterization of the GIA process. Accordingly, for a realistic water–ice redistribution during the glacial process the initial sea level has to be determined iteratively (e.g. Kendall et al., 2005).

To circumvent this problem, we consider the RSL rate, i.e. its time derivative, which depends much less on the initial state. Despite the RSL rate being a non-standard type of observation and knowing that this quantity has to be derived from a series of SLIPs resulting in increased uncertainties, we consider it as a tractable procedure. The uncertainty of RSL rates has the following relationship to RSL uncertainties:

$$\sigma_{\mathrm{rate}} = \frac{\sqrt{2}\sigma_{\mathrm{abs}}}{\Delta t}.$$

Here, the SLIP linear rate at time $t_n$ is determined from the RSL difference at $t_n$ and $t_{n-1}$. The RSL data used to compute the rates for the assimilation (which function as our observations) are taken from the reference run, $m_0$ (cf. Fig. 3). This synthetic data set is therefore well known and its statistical properties can be linked to the assimilation results.

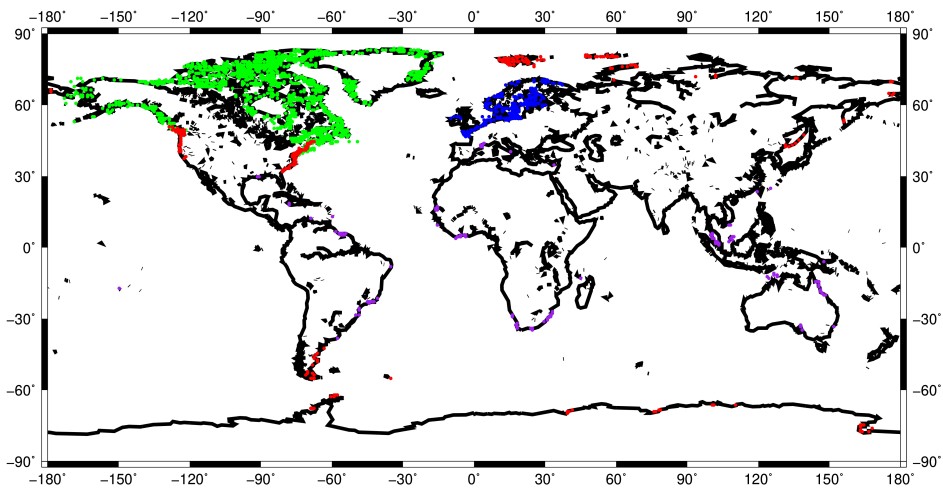

**Figure 4.** Locations of SLIPs, which are used to generate synthetic observations. The observation locations where sub-divided into four regions: Laurentide & Greenland (green), Fennoscandia (blue), Far-field (purple), and other (red).

The spatial distribution of collected SLIPs is rather heterogeneous, mainly spreading along the coasts of the continents, at islands, and concentrating to regions of large ice-water changes since the LGM. In order to run realistic scenarios, the synthetic observations were limited to locations where such data are available.

The grid point closest to each SLIP site was chosen for the representation of the synthetic data. That way, 1807 observation points were obtained. They are unevenly distributed and located mostly along the coasts of regions where large sea level changes have occurred in the past or are still ongoing, e.g., Laurentide and Fennoscandia (Fig. 4). At each location we constructed a time series of observation points, but do not consider the uneven distribution in time. Instead, we consider the points to be evenly distributed in time from LGM (last glacial maximum) to present day.

The locations of synthetic observations are based on locations of real observations in older compilations by Fleming (2000) (far-field), Art Dyke, personal comm. (Canada and Greenland), and Milne et al. (2005) for Patagonia and South America. Compilations of the US American coast are based on Engelhart and Horton (2012); Engelhart et al. (2015), and own compilations represent the Eurasian Arctic.

### 4.3 Investigated setups

This study consists of three setups. Setup 1 investigates the influence of observation uncertainty on the assimilation. In the second setup the observations were restricted to certain regions in order to test the performance of our approach when observations are not available globally. In the third setup, the time interval with available observations was restricted to after 10 ka BP until present day.

The first setup is split into two scenarios based on the Cases A–D listed in Table 2. The purpose is to investigate the influence of observation uncertainty and statistical parameters of the initial ensemble on the convergence and uncertainty of the viscosity estimates. Experiment case E was added to investigate the behaviour of the algorithm under more realistic circumstances.

In scenario one (Cases A-C), the initialisation is equal in all three cases: perturbation with noise drawn from a normal distribution $N(\mu_{\mathrm{init}}, \sigma_{\mathrm{init}}^2)$ such that $|\mu_{\mathrm{target}} - \mu_{\mathrm{init}}| = \sigma_{\mathrm{init}}$. The Cases A–C differ only in the RSL observation uncertainty. It is varied from $0.1\,\mathrm{m}$ in case A to $0.5\,\mathrm{m}$ in Case C. If $\mu_{\mathrm{init}} \leq \sigma_{\mathrm{init}}$ the target viscosity value is well covered by the ensemble PDF and the influence of different observation uncertainties can be studied.

In scenario two (Cases B and D), the influence of the initial offset (i.e. the mean of the initial perturbation, cf. Sect. 4.1) is studied. We compare test Case B (with a moderate initial offset) to a case where a large offset ($\mu_{\mathrm{init}} = 2\sigma_{\mathrm{init}}$) was chosen, such that the target viscosity value lies in a tail of the ensemble PDF. Having in mind a future application of the method to real data it is important to also reach convergence if the true value lies somewhere in a tail of the initial ensemble PDF. One must ensure that the true value is still properly sampled by the ensemble PDF. Otherwise the filter degenerates.

In case E the RSL observation uncertainties vary with time. They are set to realistic values of sea level indicator uncertainties that on average agree with values found in literature, e.g. Vacchi et al. (2018) and Khan et al. (2019). The chosen RSL uncertainties are: 7 m until 17.5 ka BP, 3.5 m for 17.5–11.5 ka BP, 1 m for 11.5–6.5 ka BP, and 0.5 m after 6.5 ka BP.

For the assimilation in setup 2, four sets of observations where compiled. In the first set, all observations where used. This gives the best possible spatial and temporal coverage. In three following scenarios, observations where restricted to 1) Laurentide and Greenland (Case LG), 2) Fennoscandia and Northern Europe (Case FS), or 3) the far-field. This was done to investigate under which conditions in the 1D model setup regional observations can be used to obtain correct global viscosity values. When considering real SLIPs, observations might be available only in certain parts of the world and it is important that our approach is proven successful under those conditions.

Looking at the temporal distribution of real SLIP observations it becomes clear that most of them date from after the last glaciation. This is due to the fact that regions showing the largest post-glacial uplift were covered by ice during the glaciation period and only after deglaciation SLIPs could form. Therefore, in setup 3 we tested our approach for the case of observations being available only after 10 ka BP. The parameters of the test in those cases correspond to Cases B10 and C10 with observation uncertainties of 0.25 m and 0.5 m, respectively (see Table 2). In this time period, RSL is mainly dominated by the Earth's deformation (post-glacial rebound) and less by changes in barystatic sea level.

In general, large ensemble sizes (especially for high-dimensional problems) are necessary to properly sample the model PDF. Due to the low dimensionality of the problem described here (only two distinct viscosity values) an ensemble size of 50 proved to be sufficient in the presented experiments.

## 5    Results

### 5.1    Consistency tests (Setup 1)

In setup 1, we studied the convergence of the weighted mean to the target values of the reference model when all available observations from the time interval 25.5 ka BP until present day are considered. Figure 5 shows the misfit measure variation
over time for scenario one (Cases A–C, cf. Table 2). The root-mean-square (RMS) of the difference between sea level rate of change observations and model predictions of each ensemble member drops quickly after the onset of the assimilation and the first resampling. The very improbable particles from the initial ensemble are disregarded already at this step. In the course of the following assimilation the RMS stays mostly below 0.5 m/ka, 1 m/ka, and 1.5 m/ka for Cases A, B, and C, respectively, and converges to values of 0.25 m/ka, 0.6 m/ka, and 0.9 m/ka towards the end of the assimilation period. In all cases, there is
a prominent RMS error peak at about 13.5 ka BP and a minor peak around 10 ka BP. These peaks are associated with larger melting phases around 14 and 12–13 ka BP.

Figure 6 shows the corresponding variation of the viscosity values. In all Cases A–C we observe very good convergence to the the upper and lower mantle viscosities of the reference model. From a state with a large variance the ensembles evolve to a state with much lower variance that is governed by the observation uncertainties and model errors. The ensemble means
converge towards the target values and stabilise within a range below $\pm5\%$ difference to the reference value for the lower mantle and $\pm2\%$ for the upper mantle. In cases B and C the ensemble spread is larger than in case A. Nevertheless, the ensemble mean is able to recover the target viscosities of the reference run within some error margin. The recovered mean values (weighted ensemble means) also lie within $\pm5\%$ and $\pm1\%$ of the target values of lower and upper mantle viscosities, respectively. The viscosity values of the upper mantle region converge more quickly than those of the lower mantle.

In scenario two (Cases B and D) we verify ensemble convergence for the case of the target viscosity value being in the tail of the initial ensemble PDF. That means we added an offset to the initial viscosities such that the target viscosities are far from the initial ensemble mean. Figure 7 shows the RMS error of the model predictions with respect to the observations for the two cases compared in this scenario. Both ensembles converge to model states that yield RMS values of about 0.6 m. The RMS variation of Case D is similar to Case B, only the initial errors are higher in Case D. The parameters of Cases B and D differ
only in the initial mean value of the ensemble perturbation $\mu_{\text{init}}$ (cf. Table 2).

Figure 8 shows the variation of the viscosities in the ensemble for the test cases of scenario two. The ensemble mean of Case D converges equally well as the one of Case B. Also, the ensemble spread is of the same order for both test cases. Therefore, if the initial ensemble's sampling density in the neighbourhood of the target value is high enough, the filter does not degenerate and the subsequent behaviour is similar to the test case with lower initial offset. In that case the weighted ensemble mean
converges to the target value.

The final experiment in setup 1 was designed to investigate the algorithm's behaviour under more realistic uncertainty conditions. In the first part of the model runs up to 17.5 ka BP the assumed observations uncertainty is very high and subsequently the algorithm convergence does not start yet. When more precise observations become available the algorithm starts to converge.

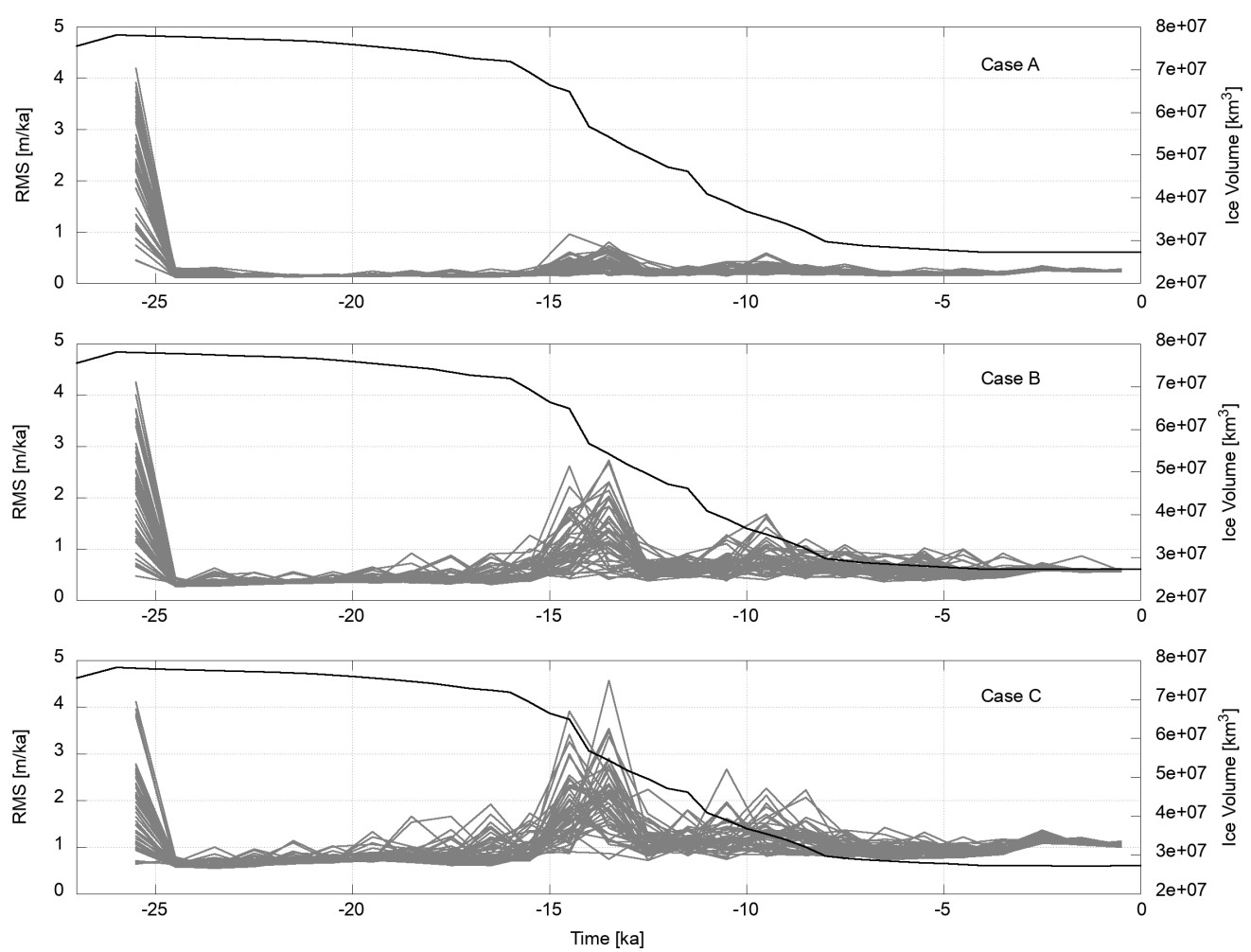

**Figure 5.** Measures of misfit variation for the ensemble of setup 1 (cases A–C) in grey. Shown are the RMS values of the difference between the sea level rate of change observations and the model predictions of each ensemble member. The black lines show the total ice volume according to the external ice model. There are spikes following large changes in total ice mass. They appear due to the large amount of fresh water that changes the RSL independently of the viscosity model.

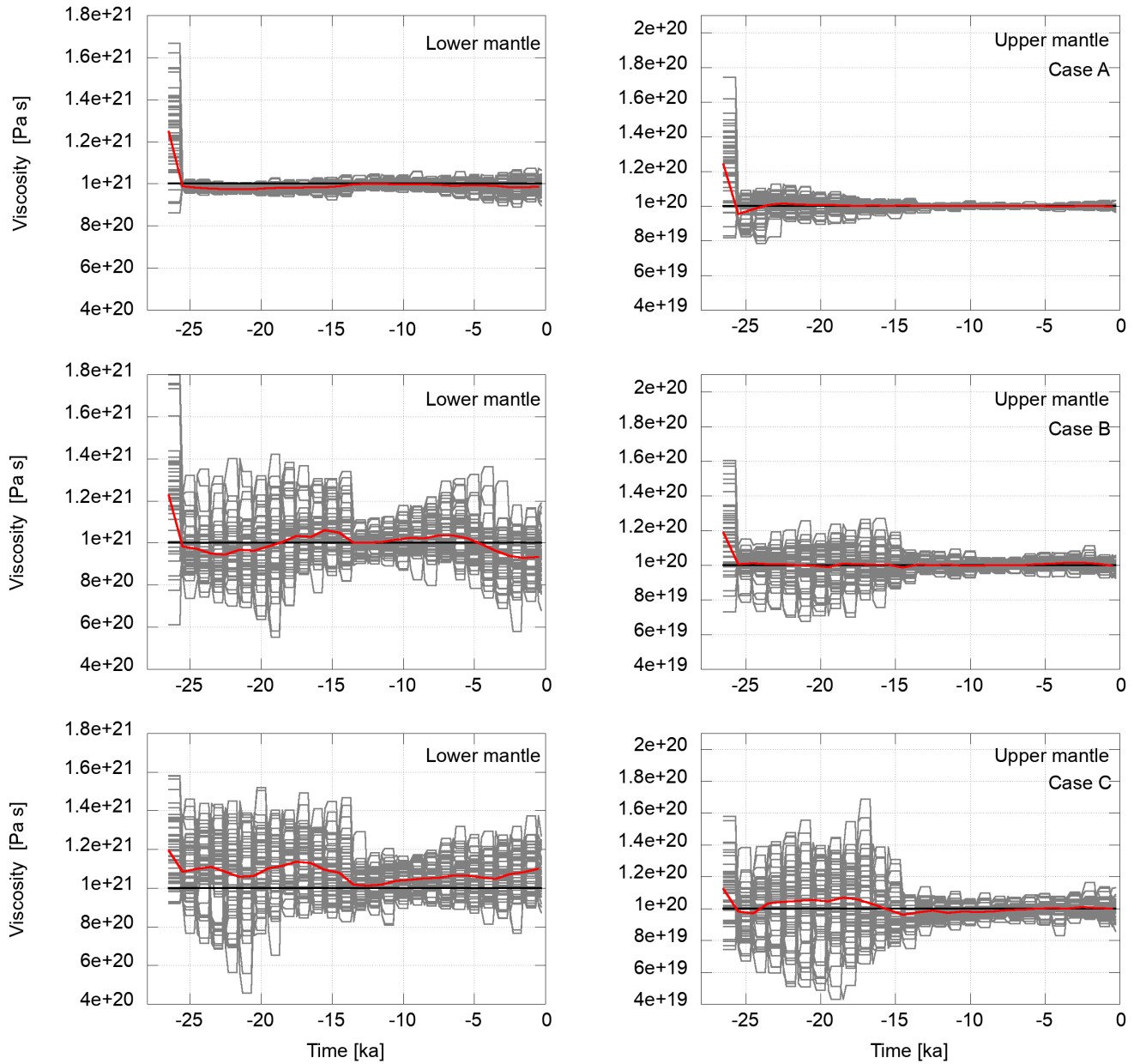

**Figure 6.** Variations of viscosity in the course of the assimilation in the lower mantle (left column) and the upper mantle (right column) for scenario one (Cases A-C, from top to bottom) in grey. The flat segments represent the viscosity values of an ensemble member during the forecast phase. When observations are available, a model state may be resampled and perturbed. This changes the viscosity values as shown. The horizontal black lines represent the viscosities of the reference experiment towards which the ensemble mean is expected to converge. The red lines are the weighted ensemble means. Ensemble members are weighted by the likelihood of the observations given the current member model state.

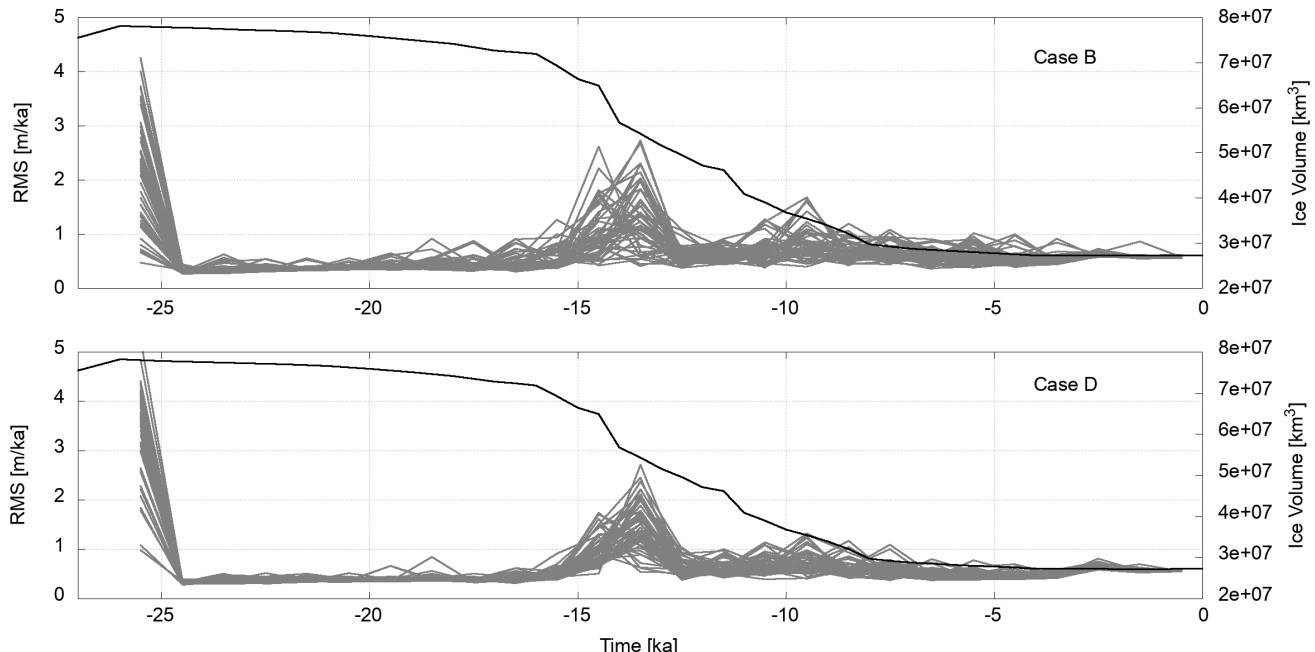

**Figure 7.** Measures of misfit variation for the ensemble of scenario two in setup 1 (Cases B and D) in grey. Shown are the RMS values of the difference between the observations and the model predictions. The black lines show the total ice volume according to the external ice model.

The variation of RMS errors of the RSL rates is shown in Fig. 9. The effect of increasing misfits following a higher melting rate are more pronounced than in cases A–D.

Figure 10 shows the variations of viscosity values in the course of the assimilation for case E. In the beginning of the assimilation there is a slight divergence of the ensemble and the ensemble mean moves away from the target values after melt water pulses. After 10 ka BP, however, there is a clear convergence for both regions, lower and upper mantle, with very good agreement of the ensemble mean with the target value for the upper mantle.

The STD of the ensemble represents the uncertainty of the parameter estimation. Figure 11 shows the STD variation for the ensemble in test cases A–D of setup 1, the situation for case E is shown in Fig. 12. The two mantle regions are shown separately since their viscosity magnitudes are very different. In both regions there are two time ranges with quite different STD levels. The first region with higher STD levels lasts until about 14.5 ka BP. After that, STD levels drop in all test cases and rise slightly towards the end of the assimilation period.

Figure 13 shows the effective ensemble size for the cases A–E. The effective ensemble size is very low after the first assimilation step for the cases with low observation uncertainties. After the first resampling and perturbation it stabilises at almost the value of the nominal ensemble size. After events of higher melting rate, the fit reduces significantly but recovers quickly to the previous level.

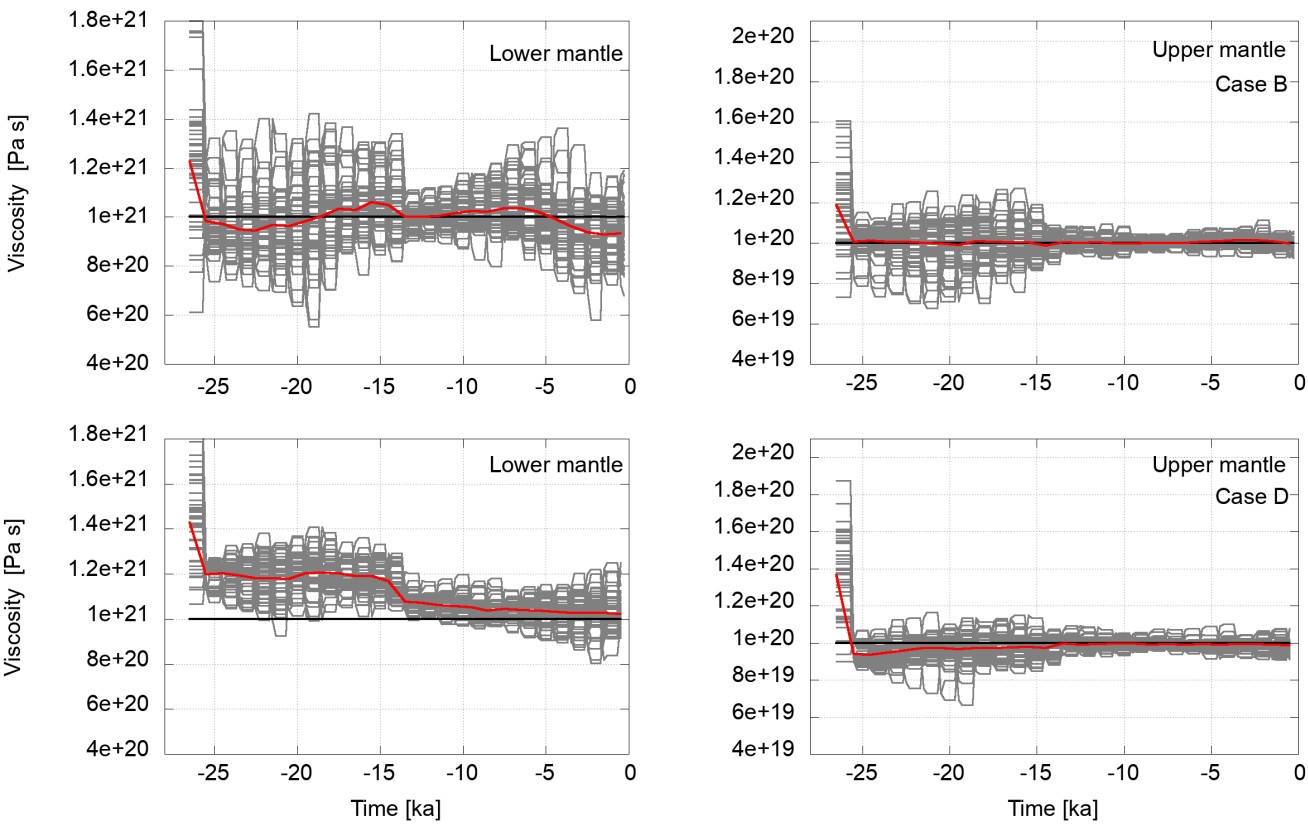

**Figure 8.** Variations of viscosity in the course of the assimilation in the lower mantle (left column) and the upper mantle (right column) for scenario two (test cases B and D, from top to bottom) in grey. The flat segments represent the viscosity values of an ensemble member during the forecast phase. When observations are available, a model state may be resampled and perturbed. This changes the viscosity values as shown. The horizontal black lines represent the viscosities of the reference experiment towards which the ensemble mean is expected to converge. The red lines are the weighted ensemble means. For the initial value all members are weighted equally. Thereafter, they are weighted by the likelihood of the observations given the current member model state.

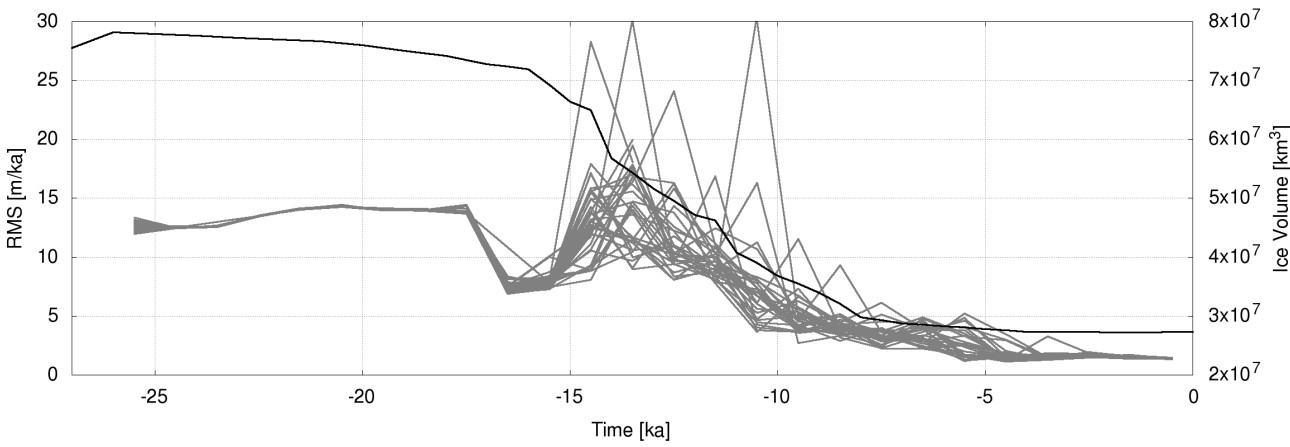

**Figure 9.** Measures of misfit variation for the ensemble of scenario three in setup 1 (Case E) in grey. Shown are the RMS values of the difference between the observations and the model predictions. The black line shows the total ice volume according to the external ice model.

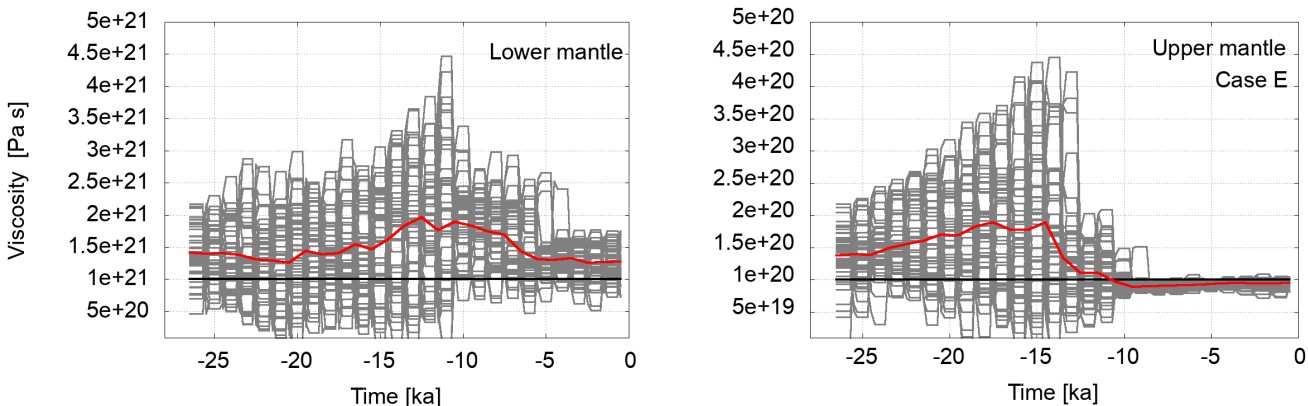

**Figure 10.** Variations of viscosity in the course of the assimilation in the lower mantle (left panel) and the upper mantle (right panel) for scenario three (test case E) in grey. The flat segments represent the viscosity values of an ensemble member during the forecast phase. When observations are available, a model state may be resampled and perturbed. This changes the viscosity values as shown. The horizontal black lines represent the viscosities of the reference experiment towards which the ensemble mean is expected to converge. The red lines are the weighted ensemble means. For the initial value all members are weighted equally. Thereafter, they are weighted by the likelihood of the observations given the current member model state. Please note the changed y-axis range in this plot.

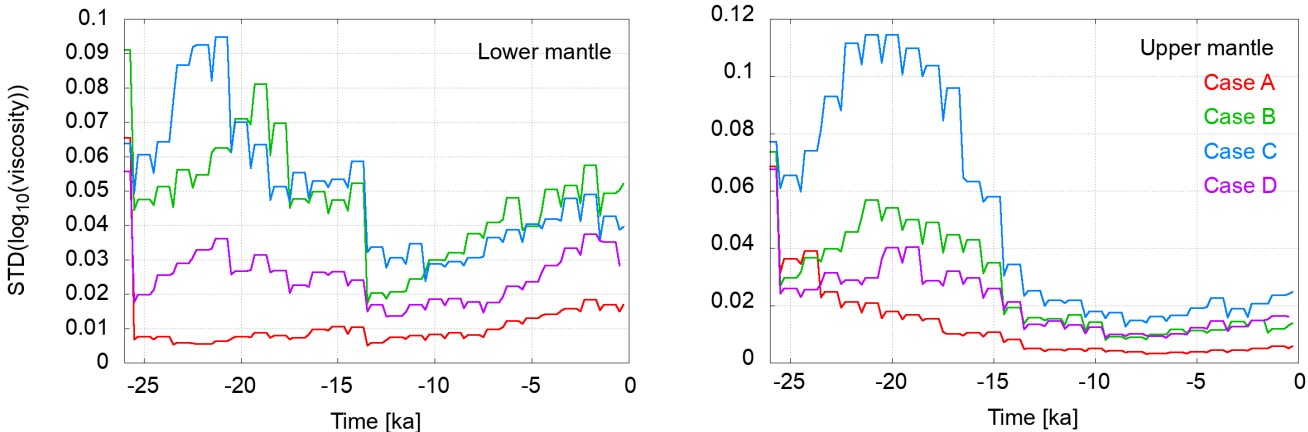

**Figure 11.** variation of the ensemble standard deviation of $\log_{10}(\nu)$ over time (red: Case A, green: Case B, blue: Case C, purple: Case D). The left panel shows the values for the lower mantle and right panel for the upper mantle, respectively. The points at the beginning and end of each plateau represent the ensemble at the beginning and end of a forecast phase. They are equal since during the forecast the viscosity remains unchanged. The drop after a plateau happens due to resampling and the subsequent rise due to perturbation (although this happens at the same point in time, the values are shifted horizontally to visualise the variation).

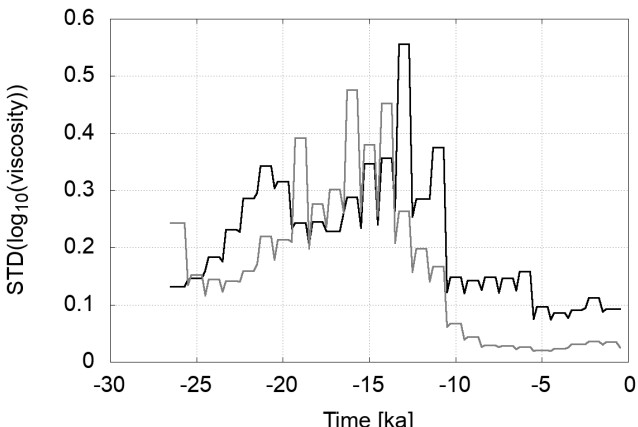

**Figure 12.** variation of the ensemble standard deviation of $\log_{10}(\nu)$ over time for case E. The black curve shows the values for the lower mantle and grey curve for the upper mantle, respectively. Due to the observation uncertainties being higher than in cases A–D the uncertainty of the viscosity estimate is also higher.

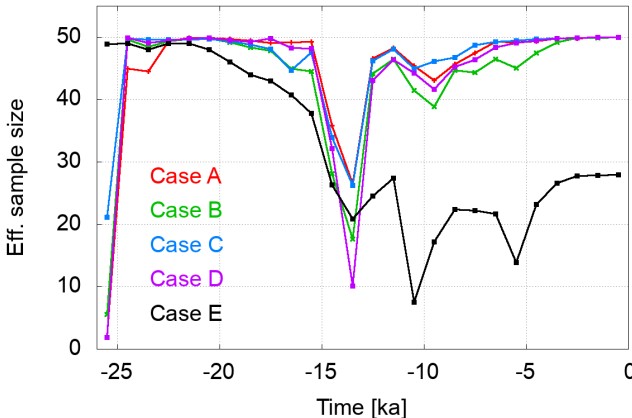

**Figure 13.** variation of effective ensemble size over time. The effective ensemble size is low in the first assimilation step if the assumed observation uncertainty is too low for the initial ensemble spread. The ratio between observation uncertainty and ensemble variance is low for case A and larger for cases B and C. It is largest for case E with very high observation uncertainties. After resampling and perturbation in the first assimilation step, the effective ensemble size almost equal to the nominal ensemble size. For cases A–D, there is a reduction of effective ensemble size after large melt water pulses at 14.5 and 9.5 ka BP, respectively. For case E the effective ensemble size reduces when the observation uncertainty is reduced significantly from one assimilation step to the next.

For case E the picture is different. The effective ensemble size is very large in the beginning and slowly decreases during the assimilation run. At assimilation steps, when the observation uncertainty is reduced significantly compared to the previous value, there is a more pronounced drop of the effective ensemble size. The lowest values are at 20% of the nominal ensemble size after changes in observation uncertainty but it rises to 50% at the end of the assimilation.

## 5.2 Regional observations (Setup 2)

In the first regional test (Case LG) we restricted the observations to those located in the area of the Laurentide ice sheet and Greenland (see Fig. 4). There are 1309 locations considered. In Case FE only 209 locations from Fennoscandia and Northern Europe are considered. The statistical parameters for the regional cases are equal to those of Case B (cf. Table 2). For this regional setup we chose locations in the near-field of large ice mass changes. Tests with data from the far-field only were not successful.

Figure 14 shows the variation of the RMS error for the RSL observations for the Cases LG and FS. The RMS variation is similar to the variation in setup 1, where the full data set is applied (see top panel of Fig. 7). In case of the complete data set, the final RMS values at present day are about 0.6 m. This also holds for Case LG. The Fennoscandian data set with considerable fewer observations distributed over a smaller area shows slightly larger RMS values for the peaks at 15 ka and 9 ka BP and also at present day (about 0.9 m).

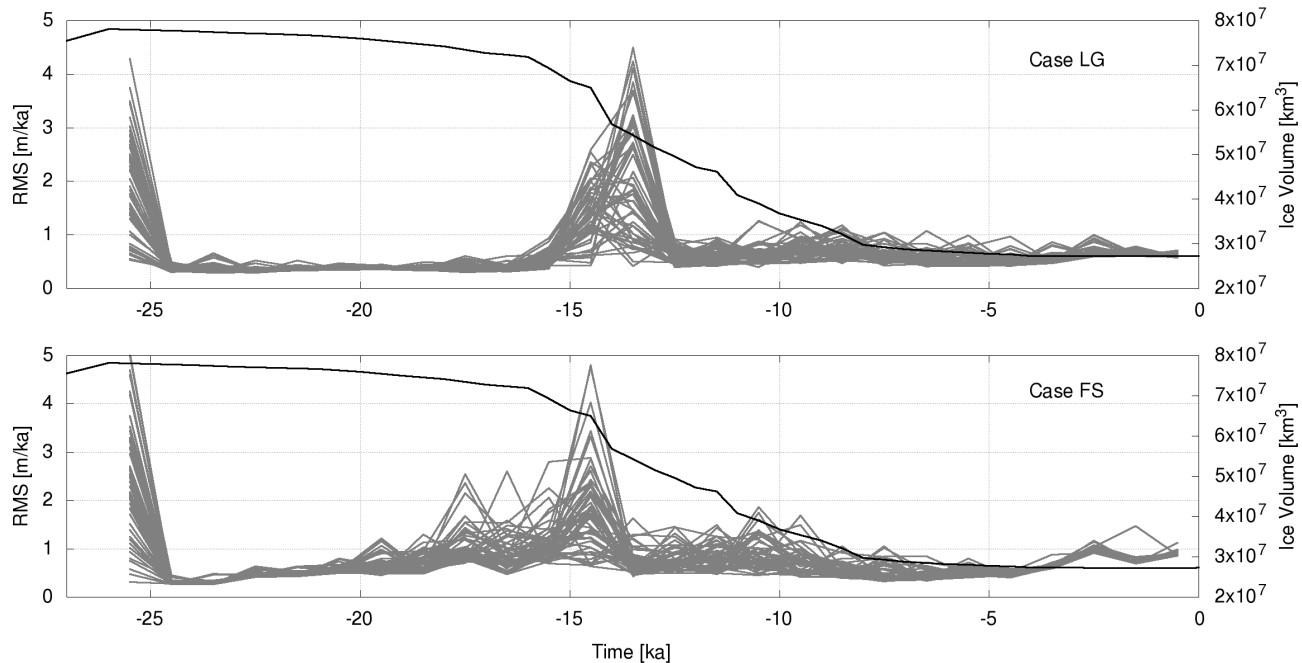

**Figure 14.** Measures of misfit variation for the ensemble of two regional observation sets: Northern America and Greenland (top) and Fennoscandia (bottom). Shown are the RMS values of the difference between the observations and the model predictions. The black lines show the total ice volume according to the external ice model. There are spikes following large changes in total ice mass that can be explained by different response times of the reference model and the ensemble member to changing mass load. The response time depends on the mantle viscosity of the individual model.

Figure 15 shows the model state variation of the viscosity values of each ensemble member and the weighted mean. Clearly,
the weighted means converge towards the target values. The convergence of lower mantle viscosity is slower and shows more
variability than the upper mantle viscosities. For the North America and Greenland region, the variation over time shows the
same characteristics as in the complete data set experiments. There is a period with large variability until about 13.5 ka BP
after which it drops significantly. In case of Fennoscandia, there is no such drop-off in variance (see also Fig. 16) in the lower
mantle viscosities. It is present, though less prominent, in the upper mantle viscosities. So, we can confirm that the lower
mantle viscosity cannot be resolved by the uplift characteristics of the smaller Fennoscandian ice sheet.

## 5.3   Time interval tests (Setup 3)

In setup 3 only observations taken after 10 ka BP where used in the assimilation. This corresponds to times after the last
deglaciation. With this setup we demonstrate that the algorithm can reach convergence in a short period of time that is very
relevant for real observations as most SLIPs originate from the period since early Holocene.

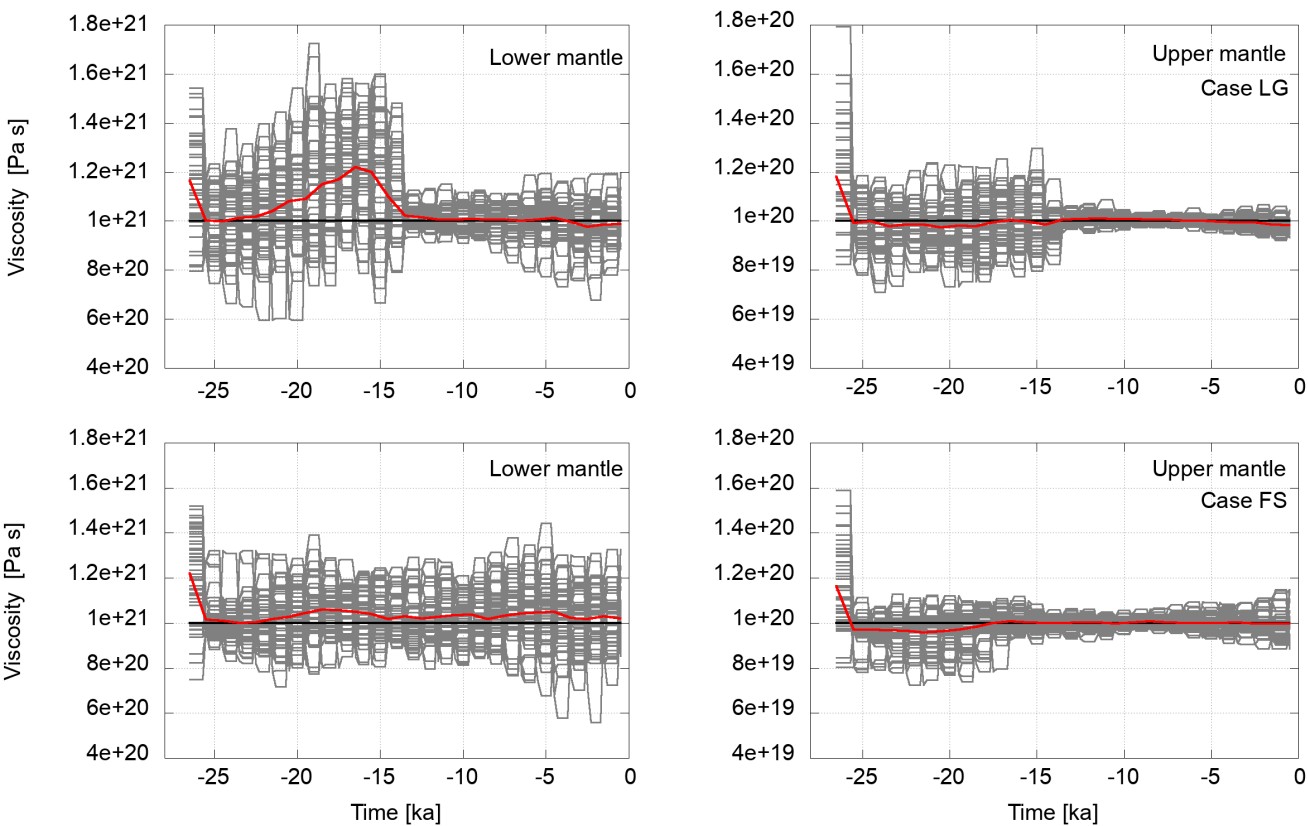

**Figure 15.** Variations of viscosity in the course of the assimilation in the lower mantle (left column) and the upper mantle (right column) for test cases with regional data sets: Northern America and Greenland (top row) and Fennoscandia (bottom row). The flat segments represent the viscosity values of an ensemble member during the forecast phase. When observations are available, a model state may be resampled and perturbed. This changes the viscosity values as shown. The horizontal black lines represent the viscosities of the reference experiment towards which the ensemble mean is expected to converge. The red lines are the weighted ensemble means. For the initial value all members are weighted equally. Thereafter, they are weighted by the likelihood of the observations given the current member model state.

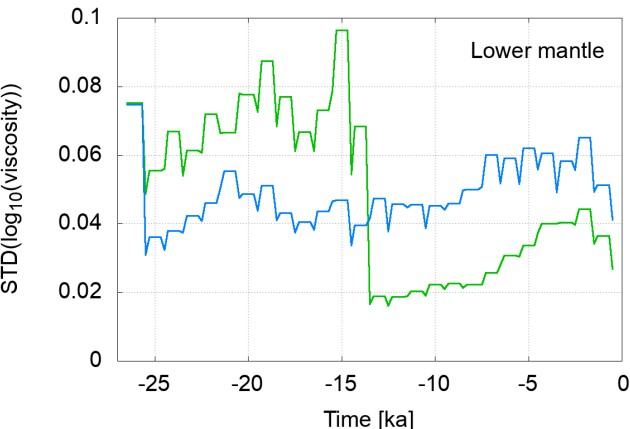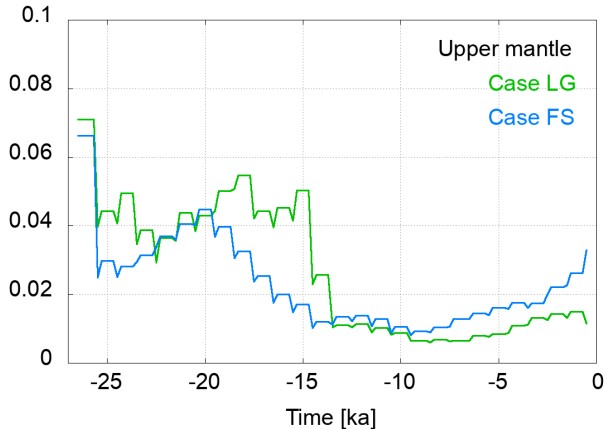

**Figure 16.** variation of the ensemble standard deviation of $\log_{10}(\nu)$ over time (green: Laurentide & Greenland, blue: Fennoscandia). The left panel show the values for the lower mantle and right panel for the upper mantle, respectively. The points at the beginning and end of each plateau represent the ensemble at the beginning and end of a forecast phase. They are equal since during the forecast the viscosity remains unchanged. The drop after a plateau happens due to resampling and the subsequent rise due to perturbation (although this happens at the same point in time, the values are shifted horizontally to visualise the variation).

Figure 17 shows the variation of the RMS misfit of RSL for setup 3. The final RMS values are in the same range (0.5 m for Case B10 and 1 m for Case C10, respectively) as the values for the corresponding cases considering observations from 25.5 ka BP to present day. There is little variability within the ensembles. There are no RMS spikes in this time period.

Figure 18 shows the variations of viscosity values in the course of the assimilation over time. The convergence of the weighted mean to the target values is very fast. Although there is some variability within the ensemble the weighted mean is very stable over time.

Figure 19 shows the variation of the ensemble's viscosity STD over time. There is a quick drop-off after assimilation onset. Thereafter, the STD stays fairly constant until the end of the assimilation. Only in Case C10 (global distribution since 10 ka BP with larger uncertainties) a slight rise of STD is visible in the upper mantle. We see the same features in the STD in setup 1 where the entire data set was used (see Fig. 11) but the variability is slightly lower in this 10 kyr case.

# 6 Discussion

## 6.1 Setup 1

The results of setup 1 show that the weighted ensemble mean converges to the target values during the assimilation. The final misfit of RSL and the ensemble variance scales with the assumed observation uncertainty. This is expected since with increasing observation uncertainty the correction of the dynamic models in the assimilation step is reduced. In the PF we used,

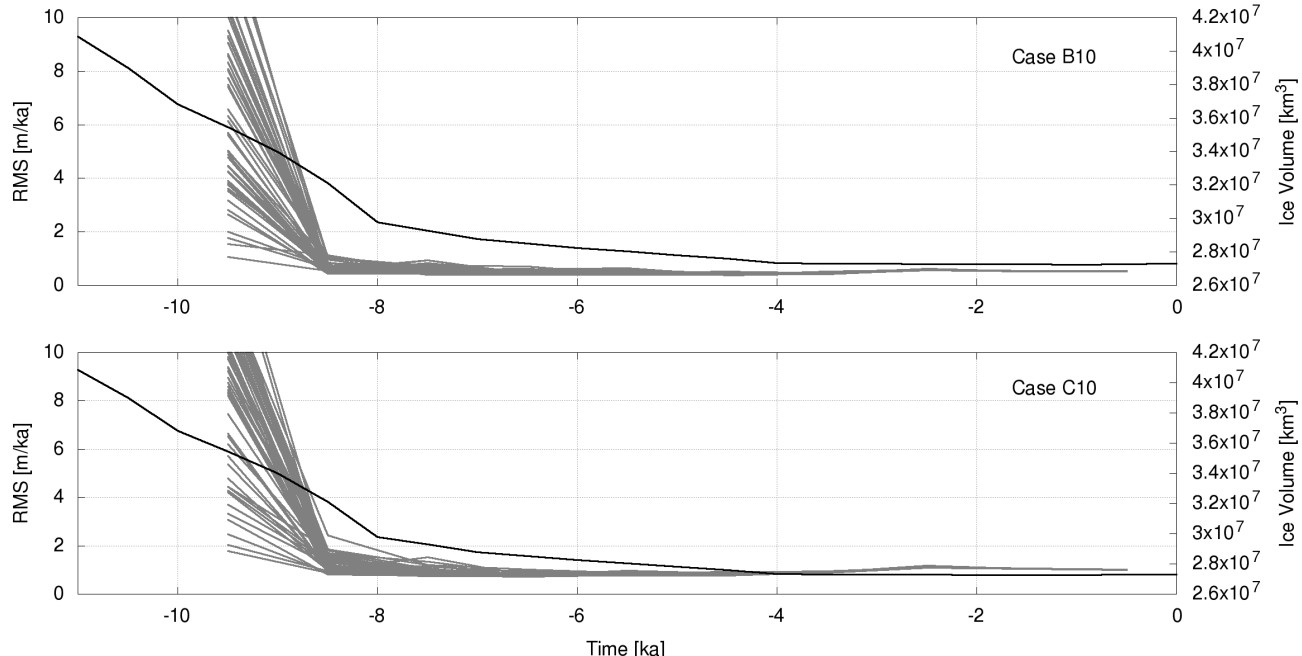

**Figure 17.** Measures of misfit variation for the ensemble of two sets of observations dating from after 10 ka BP. The statistical parameters of Cases B10 and C10 are equal to those of Cases B and C in Table 2. The assumed observation uncertainties are 0.25 m (top) and 0.5 m (bottom) Shown are the RMS values of the difference between the observations and the model predictions. The black lines show the total ice volume according to the external ice model.

ensemble members with low likelihood are resampled to model states with high likelihood. Larger observation uncertainties reduce the separability of models based on that measure. As a consequence fewer models are resampled to better model states, the convergence slows down and the final ensemble shows a larger variability.

Although in general the convergence of the ensemble is very good, there are some peaks in the RMS error of RSL rates at about 13.5 ka BP and 10 ka BP that appear suddenly and slow down the convergence. These peaks coincide with larger changes in ice volume (melt water pulses) with a delay of 1 to 2 kyrs. They can be explained by the large amount of melt water flowing into the oceans and resulting in an RSL signal that dominates the sea level change at those times. However, that signal is independent of the viscosity model. As a consequence, the ensemble members cannot be effectively evaluated based on the current RSL rate misfit. Models that are relatively far from the ground truth are able to stay in the ensemble and produce large misfits in the first evaluation step after the melt water pulse. In the following time steps with reduced melt rate, the mantle rheology again dominates the RSL rate, and the RMS errors are reduced to previous levels. The general level of misfit, as well as the peaks after sudden changes in ice volume, scale with the observation uncertainty assumed in that specific test case. This result shows, the interplay between melt water and the Earth's response hinders the inference of structural parameters during this phase as the barystatic sea level change dominates, which is independent of the Earth's structure.

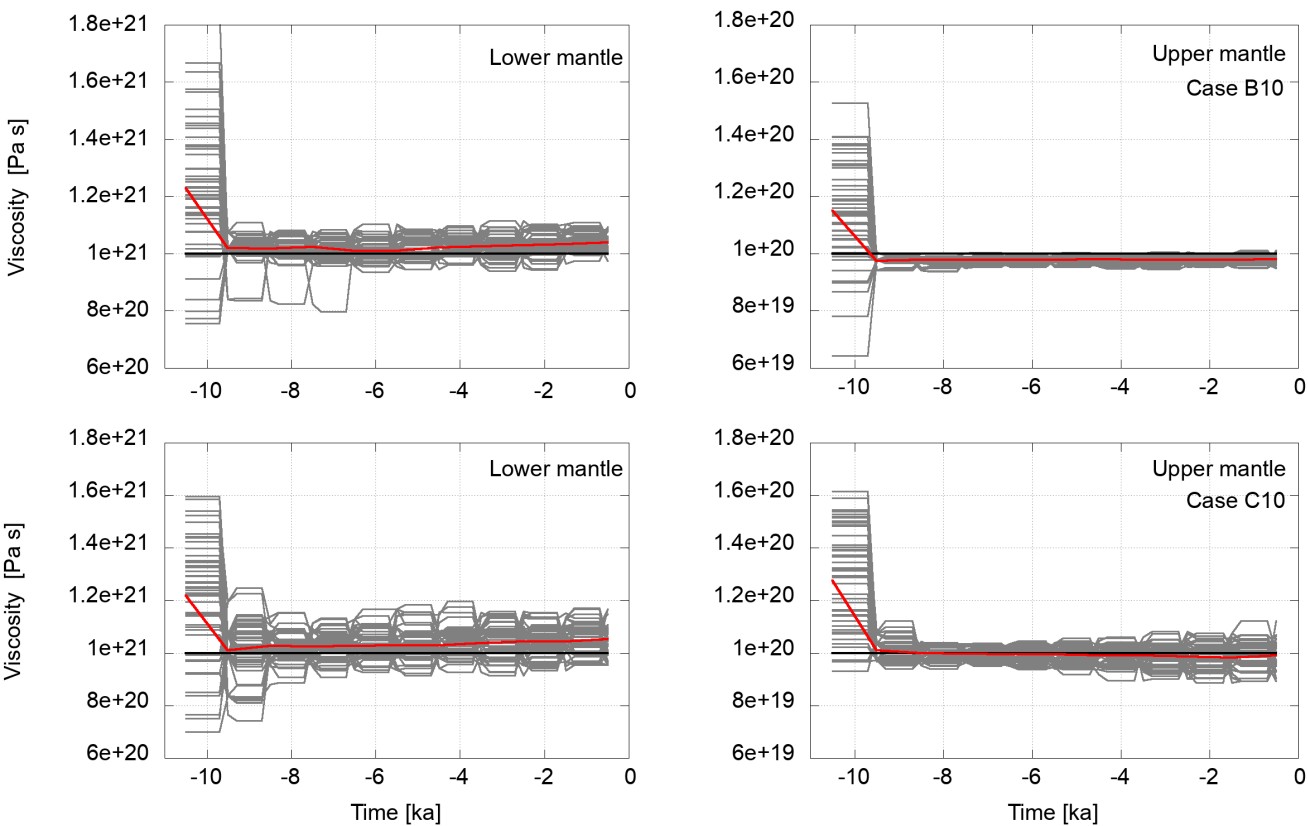

**Figure 18.** Variations of viscosity in the course of the assimilation in the lower mantle (top) and the upper mantle (bottom) for test cases with observations starting from 10 ka BP. Observation uncertainties are 0.25 m (left column) and 0.5 m (right column). The flat segments represent the viscosity values of an ensemble member during the forecast phase. When observations are available, a model state may be resampled and perturbed. This changes the viscosity values as shown. The horizontal black lines represent the viscosities of the reference experiment towards which the ensemble mean is expected to converge. The red lines are the weighted ensemble means. For the initial value all members are weighted equally. Thereafter, they are weighted by the likelihood of the observations given the current member model state.

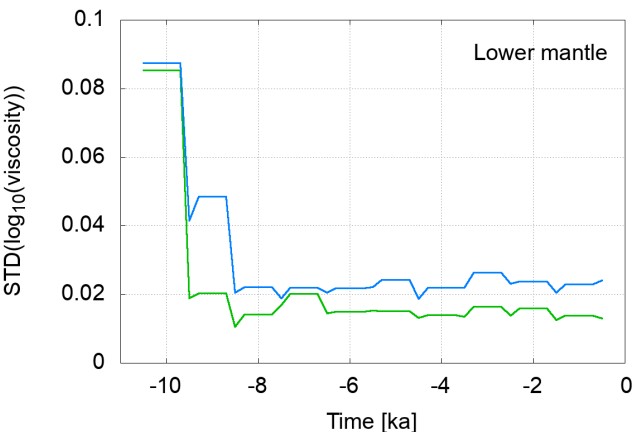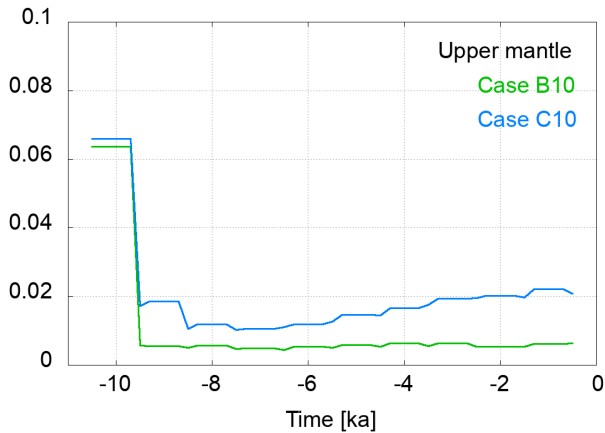

**Figure 19.** variation of the ensemble standard deviation of $\log_{10}(\nu)$ over time for the 10 kyr assimilation (green: Case B10, blue: Case C10). The left panel shows the values for the lower mantle and right panel for the upper mantle, respectively. The points at the beginning and end of each plateau represent the ensemble at the beginning and end of a forecast phase. They are equal since during the forecast the viscosity remains unchanged. The drop after a plateau happens due to resampling and the subsequent rise due to perturbation (although this happens at the same point in time, the values are shifted horizontally to visualise the variation).

The convergence of the viscosities to the target values is very good. Generally, the convergence decreases slightly from Case
A to C. The larger observation uncertainties allow particles to survive which are farther from the target model. The viscosity in the lower mantle show slower convergence than in the upper mantle. This is a general problem in GIA modelling (e.g. Caron et al., 2018). The reason for that is that viscosity changes in the lower mantle take more time to have impact on sea levels. In general, small deformations at the surface have little impact on lower mantle deformation and with increasing depth it becomes more difficult to constrain mantle viscosity by surface deformation. This is also apparent when looking at the variability within
the ensemble as shown in Fig. 11. The slightly rising variability towards the end of the assimilation period might be due to the low magnitude of RSL rates in younger times. With very small signals it is difficult to correct the models properly and the variance introduced by the perturbation leads to a slight rise in variance within the ensemble.

The convergence of the ensemble mean viscosities to the target values of the reference runs in the presented cases show that with our approach we are able to recover mantle viscosities within a reasonable uncertainty range. This is even the case if the
355 initial ensemble's PDF is far from the target values, i.e. the target value is in one of the tails of the initial PDF. A requirement is, however, that the sampling density of the ensemble near the target value is still high enough such that the filter does not degenerate. Furthermore, the convergence is strongly influenced by the assumed observation uncertainties. Large uncertainties on the one hand slow down the convergence and lead to larger final variance within the ensemble. On the other hand they reduce the chance of degeneration since particles with larger deviations from the target values are assigned higher likelihoods
if observation uncertainties are higher.

Test case E was run with considerably higher assumed observation uncertainties. The aim was to check the algorithm's success under near-realistic circumstances. That involves high uncertainties for older observations and step-wise uncertainty decrease as observations become more recent. Under those conditions, the ensemble converges only after some time, when the uncertainty has become lower than 7m. Clearly, the initial observation uncertainty results in an ensemble variance that is larger than the variance of the initial ensemble. Only after observations with lower uncertainty are available, the ensemble converges towards the target values. Prior to that, also members relatively far from the ground truth are assigned likelihood values that allow them to survive and the ensemble mean does not yet converge towards the target value.

## 6.2 Setup 2

The results of setup 2 show that we are able to recover target viscosities with only a subset of available observations. The uncertainty of the final estimations seems to depend only little on the size of the observation region. The 3-layer model with two mantle layers is simple enough to recover the viscosities with only little more than 200 observation locations. However, there are differences in the lower mantle viscosity estimation. The larger variance in lower mantle viscosity from Fennoscandia when compared to Laurentide can be explained by the smaller size of the Fennoscandian ice sheet. Accordingly, the GIA of Fennoscandia is less sensitive to lower mantle viscosity structure (e.g. Mitrovica and Peltier, 1993; Lambeck et al., 1998). For the upper mantle there seems to be only little difference in the viscosity variance between the two cases. For both regions, signals from their respective ice shields are large enough to constrain the Earth structure.

## 6.3 Setup 3

In setup 3 we show that it is possible to estimate the target viscosities also when only observations from a short time interval, i.e., from 10 ka BP until present day are available. The reasoning behind those test cases is that the 10 ka BP marks the end of the last deglaciation and most real observations date from after that. The RMS misfit of RSL rates obtained in these tests drop quickly after the onset of the assimilation. At 8 ka BP, melting of Laurentide and Fennoscandian ice shields is finished and from there on, ice mass changes can be seen only in Greenland and Antarctica. Therefore, in the time interval investigated here, the dominating ongoing process is the post-glacial rebound. All model realisations were able to reproduce that (relatively slow) GIA processes taking place and there is little variance in the ensemble. Such behaviour was already observed in setups 1 and 2 where the variance also drops significantly after 10 ka BP.

The results of this experiment show that our algorithm can quickly converge to the target values of the reference model under quiet conditions. With "quiet" we mean there are no large changes in global ice mass within a short period of time and therefore the models have enough time to react viscoelastically to the new mass load. In that case, the RSL development is strictly a function of viscosity describing an exponential decrease (the relaxation process). On the other hand, if larger ice mass changes are present, the algorithm also converges but it takes longer and several assimilation steps are needed until the models adapt to the new mass load.

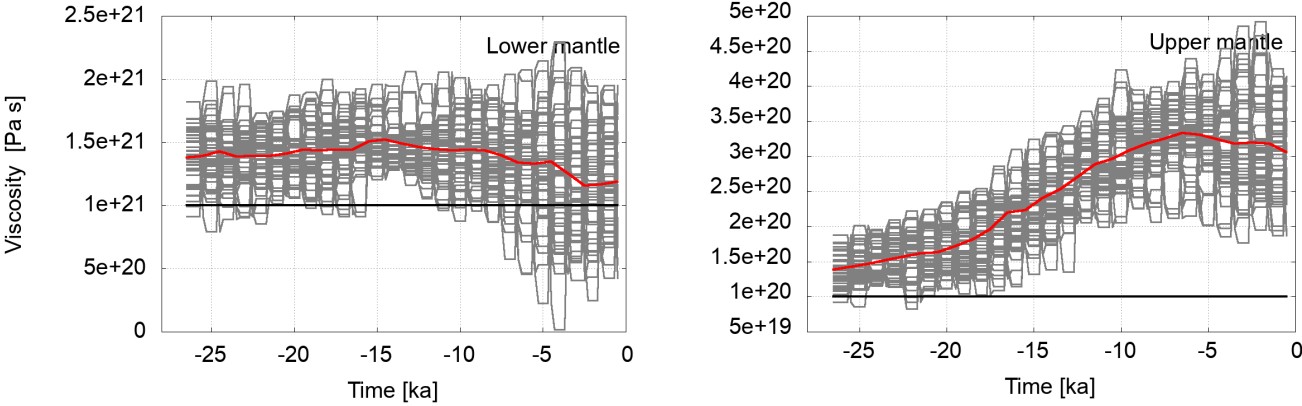

**Figure 20.** Development of ensemble with erroneous ice load. The ice load was increased globally by 5%. Red lines denotes the ensemble means, black lines the target values.

## 6.4 Ice load

Ice models are a source of large uncertainties in GIA modelling. Usually, no uncertainties are provided for global ice models. However, ice histories from different approaches, e.g. ICE-5G by Peltier (2004), ICE-6G by Peltier et al. (2015), PaleoMist 10 by Gowan et al. (2021) or NAICE by Gowan et al. (2016), reveal large deviations in ice thickness and extension during deglaciation. This indicates that the uncertainties of ice models are rather large. Using different ice models in a GIA study leads to obtaining different Earth rheology parameters (Zhao et al., 2012). The history of surface mass load strongly influences the mantle viscosity values obtained in our approach. Hence, the correctness of viscosity values obtained with real observations strongly depends on the correctness of the ice model. In our study we observed that relatively small changes of ice mass load lead to quite large changes of the ensemble mean. The lower mantle is less affected by the distorted mass load than the upper mantle. For the latter, the viscosity estimations are larger than the target value since only a more viscous mantle can explain the observed sea level variations given the increased ice load. Figure 20 shows the development of viscosity during the assimilation and the divergence from the initial values, where the ice thickness was increased globally by 5%. We had to reduce the integration time step from 20 to 5 years to stabilise the forward calculations under such erroneous ice load conditions. With that done, the effective ensemble size was mostly greater than 47 with only one drop below 43. The ensemble means show large deviations from the target values. The increased ice load leads to larger viscosity estimations since only a more viscous mantle can explain the observed sea level variations given the increased ice load. But the large ensemble variance underlines that this effect does not have the same impact on all locations. This example shows that our approach allows to evaluate which Earth structure is needed to "correct" differences in ice histories and reveals which parameters of the Earth structure are mainly affected. But keep in mind that also other combinations of ice history and Earth structure are possible to achieve same sea level reconstructions (non-uniqueness in GIA modelling, (Whitehouse, 2018)).

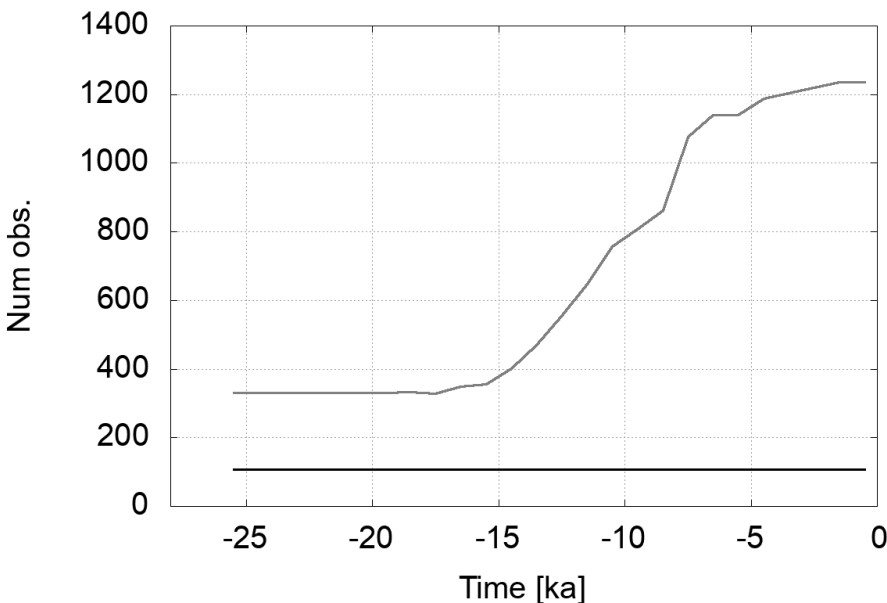

**Figure 21.** Number of observations in ice-free locations over time. The grey line shows the development of number of observations. The horizontal black line is the number of far-field observations. Far-field observations do not contribute to constraining mantle viscosities.

## 6.5 Temporal observation distribution

The focus of GIA related sea level research lies on reconstructions of the deglaciation since the last glacial maximum (e.g., Van de Plassche, 1986; Düsterhus et al., 2016; Carlson et al., 2019), which constrains the main range of available sea level data to be younger than 20 ka BP. While we considered a spatial distribution based on available data sets (cf. Unger et al., 2012), we assumed to have observations every 1 kyr for the whole assimilation period. The amount of available sea level data, however, peaks during late Holocene at $\sim$ 8 to 12 ka BP and is considerably smaller for earlier periods (e.g. Rosentau et al., 2021). A further test, where only those observation locations were considered that were ice free, showed that during the early assimilation period with large ice covers, the number of near field observations is too small to effectively constrain mantle viscosities. Figure 21 shows the number of observations at ice-free locations for the global dataset. Prior to, say, 12 ka BP there are not enough observations to effectively constrain mantle viscosities. Nevertheless, towards the end of deglaciation the number of observations becomes large enough to obtain well constrained mantle viscosities. This is in agreement with the experiments of setup 3 starting at 10 ka BP.

In order to compute RSL rates from the RSL observations, several data points are necessary at one location. From the available sites which are commonly used to reconstruct the temporal evolution of the sea level from the late Pleistocene or Holocene to present day, about 20 to 30% contain more than 10 samples. This, of course, further reduces the number of

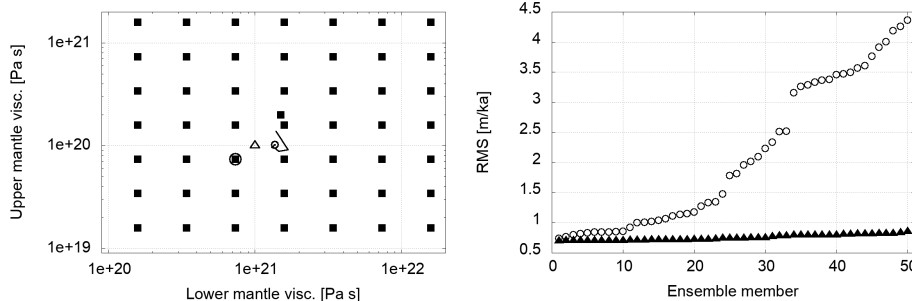

**Figure 22.** Left panel: viscosities of the (initial) ensemble for the comparison between assimilation and classical approach (solid squares), final mean state of assimilation approach (small circle) and target values (triangle). The black line is the path of the mean state during the assimilation after each resampling step. The larger circle denotes the member of the classical approach ensemble with lowest RMS error. Right panel: RMS errors of RSL rates of change for the ensemble of the classical approach (circles) and PF approach (triangles).

observations. However, from setup 2 we have seen that also with a limited number of observation sites (e.g. 209 in case of Fennoscandia) it is possible to recover the mantle viscosity target values.

## 6.6 Other influences

In this study we present a somewhat idealised setup that is used for the development of the approach. There are several factors and ambiguities that play a role that could either not be addressed in the scope of this study or are beyond the current ability of the approach. One significant parameter in GIA is lithospheric thickness. Firstly, it strongly varies laterally while we only assume a 1D Earth model. Secondly, there is a trade-off between assumed lithospheric thickness and obtained mantle viscosity values (Bergstrand et al., 2005). Hence, an erroneous assumption of lithospheric thickness can lead to errors in mantle

viscosity estimations. However, one can run several assimilations with different thickness values to obtain an estimate of the correct value. Also with a long enough observation time series the different time scales of lithospheric elastic deformation and viscous mantle deformation might be solved.

## 6.7 Comparison to classical approach

In order to estimate the results of the PF approach, we made a simple comparison to the so-called classical approach. In the

440 latter, an ensemble of models with fixed viscosity values is propagated in time and the resulting sea level rates of change are computed. At the end of the model run period RMS errors of the predicted sea level rates of change are calculated. Here, the RSL rate misfits are compared to the misfits of rates obtained with the PF approach. We constructed an ensemble of 50 models that was used as the starting ensemble for both methods. The ensemble is shown in Fig. 22 (left panel). For the classical approach we ran the ensemble from 10.5 ka BP until 0.5 ka BP and calculated the RMS error of the RSL rates obtained from

445 the models given the (synthetic) observations at 0.5 ka BP. The observations (disturbed by observation noise) were the same that were used for the assimilation at the final step at 0.5 ka BP.

The RMS errors of the ensemble in the classical approach ranged from a maximum of 4.36 m/ka to a minimum of 0.74 m/ka. The PF approach yielded misfits between 0.86 m/ka and 0.70 m/ka. In the latter case, the misfit is mostly due to difficulties in lower mantle viscosity determination. The RMS errors of each ensemble member are shown in the right panel of Fig. 22. The spread in the final state of the PF ensemble is much lower than in case of the classical approach and there is strong confidence in the correctness of the result. In the case of the classical approach the best model is not well distinguishable from the few next best models. This lowers the confidence in the result in this case.

The computation times of each approach are in the same order. It took 61 minutes for classical approach and 64 minutes for the PF approach. The relative overhead of the PF is smaller in this case than in the scenarios described in Sect. 4 since the integration step was reduced to 5 years in this comparison. This was necessary for the PF approach due to the large initial perturbation of the ensemble.

If one member of the initial guess ensemble in the classical approach is very close to the true values, the resulting RMS error can be very small and outperform the PF approach. However, the PF approach yields a result that is as close to the true values as the observation uncertainty permits.

## 7    Conclusions

We have shown that our algorithm is able to recover a synthetic mantle viscosity structure through assimilation of paleo sea level rates of change. This is the case even if the target viscosities are located in a tail of the initial distribution, thereby reducing the need for a very accurate initial guess. Furthermore, in contrast to the classical approach, the viscosity values obtained as the final result need not to be part of the initial ensemble. This has the potential to lead to viscosity values that are closer to the truth than any member of the initial ensemble. This is even more important when the number of mantle layers is increased and due to the larger number of possible combinations it becomes more difficult to cover the whole space of parameter combinations in the classical approach.

Another advantage of our approach is the intermediate results that are obtained in every assimilation step. The behaviour of RMS errors with time points to specific events that are difficult to handle play an important role in the modelling process. Those are e.g. the melt water pulses. Also, the possibility to constrain mantle parameters with the observations at a given point in time can be investigated. This depends on the number of observations available at a certain time span and on their distribution. With only observations from the far-field, e.g., we were not able to recover the GIA processes and the governing mantle viscosities. This is due to the eustatic sea level change controlling the signal in those regions (Steffen and Kaufmann, 2005). As during times of large ice covers mostly only far-field data are available this limits the applicability of our approach during strong glaciation. However, the experiments with only a short observation period from 10 ka BP until present have shown that such a limited time with high quality observations in the near field is sufficient to obtain the target viscosity values.

Some of the assumptions made about observation uncertainties need closer inspection. Real SLIP uncertainties are usually in the range of 0.5 m to 1 m for stratigraphic data and as precise as 0.1 m/ka to 0.2 m/ka for a few very exact data points only (depending on the stratigraphic regime and the age of the SLIP) (Shennan et al., 2015, pp. 3–25). With the exception of

case E, we considered RSL uncertainties of 0.25 m to 0.5 m for all observations. Firstly, this seemed to be a good compromise given the range of real SLIP uncertainties. Secondly, we wanted to study several influences on the convergence of the ensemble that should not be masked by large model deviations originating from large observation uncertainties. However, in case E we showed that also with more realistic observation uncertainties the target values can be recovered.

In addition to SLIPs, which are defined as band limits of paleo sea level, other sea level data only indicate an upper or lower bound of paleo sea level (e.g. Khan et al., 2015). Accordingly, we have to extend the error handling towards non-Gaussian error distributions (e.g. Hibbert et al., 2016; Latinović, 2021) when incorporating such observations in the future.

When applying the approach to real observations, it will be most promising if observations from after substantial deglaciation are used. This ensures a sufficient number of data points to constrain mantle viscosities and we have shown to be successful in that setup.

The ensemble size was limited to 50 members, mainly due to computational costs. For a proof of concept with synthetic data this ensemble size is sufficient. In order to recover real viscosities and as target values are unknown, a larger ensemble sampling a wider range of viscosity values might be necessary. On the other hand, when starting with large observation uncertainties that are gradually reduced as observations become more recent, a wide range of parameters can be sampled without the danger of filter degeneracy.

The effective ensemble size which is a measure for the robustness of the ensemble is very close to the nominal ensemble size in most of the cases. This indicates the the filter is far from degeneration. Only in case E, where realistic observation uncertainties were assumed, the effective ensemble size is reduced to about 50% towards the end of the assimilation. Stronger drops in effective ensemble size appear at times when the observation uncertainty is reduced from one assimilation step to the next. However, the ensemble recovers from that reduction within a few assimilation steps. A smoother transition from high to low observation uncertainties could help to reduce this effect.

The computational costs are higher than for a pure forward model run with the same ensemble size. This is due to the overhead of the particle filter. The amount of overhead depends on the ensemble size since some parts of the filter are performed in serial mode. With our ensemble size the overhead was about 30% to 40%. The exact values depend on the allocation of compute nodes in the cluster.

Another crucial point is that we compute rates of sea level change from two observations at a given location. The linear rate is attributed to the younger boundary of the time interval while it really is a mean value for the whole interval. This introduces errors in time of observation and magnitude of sea level change. This is not a problem in our twin-experiment since observations and model predictions are treated the same way. However, if real SLIP data are used, there are additional dating errors for the sea level estimates. They have an impact on the rate of change uncertainty and their consequential influence on the uncertainty of the viscosity estimates is a point future investigations need to pay attention to. The motivation for using sea level rates of change was the fact that in the applied assimilation method over time, we cannot iteratively determine the initial topography. With rates we were able to overcome that problem, however, they have other disadvantages. For example, the number of locations with repeated observations (where rates can be computed) is much smaller than the total number of observation locations (about 30% for the considered data). This limits the quality of the viscosity estimation. But other

observation types are possible, e.g. differences to a defined base level. This has advantages of both, absolute sea level values and rates.

Although RSL rates are a non-standard type of observations there are studies that model RSL rates during the Holocene, e.g. Khan et al. (2015). The uncertainties given by Khan et al. (2015) for the near-field range from 1.0 m/ka to 3.4 m/ka around 10 ka BP. For the most recent time interval of 2 ka BP until present day they give uncertainties of 0.2 to 0.4 m/ka at four locations at the coasts of North America and one location with 1.4 /ka. The uncertainties for the observation point in Greenland are slightly higher. They range from 2 m/ka at 10 ka BP to 1.0 m/ka at present day. Uncertainty values for the three North European sites range from 0.5 m/ka to 1.3 m/ka around 10 ka BP and from 0.6 m/ka to 1.9 m/ka for the most recent period. This is in agreement with the uncertainties we assumed in case E.

In this first approach we cover a rather simple 1D Earth structure with two mantle layers of constant viscosity and did not consider uncertainty in lithosphere thickness. While this is helpful for the algorithm development, more realistic scenarios involve radial viscosity profiles and even 3D viscosity variations. For a profound impact on viscosity parameter estimation and regional sea level changes this is an issue that will be addressed in future work.

Available SLIPs are sparse in the distant past and become more numerous as recent time is approached. But the situation improves due to various groups working on constraining paleo sea level rise under PAGES like HOLSEA (Khan et al., 2019) and PALSEA (Carlson et al., 2019).

Nowadays, even more sea level and ice-mass data based on GPS, tide gauges or measurements of mass redistribution with satellite missions such as GRACE and GRACE-FO are available. Uplift rates from GPS measurements are widely used to invert mantle properties (e.g. Argus et al., 2021; Zhao et al., 2012; Bergstrand et al., 2005). Paulson et al. (2007) showed that post-glacial rebound data alone cannot resolve more than two viscosity layers without suffering from trade-off between neighbouring layers. Caron et al. (2018) showed that especially in polar regions, the uncertainty of deformation trends computed from 14 years of GRACE data is lower than GIA uncertainty. Therefore, additional GNSS observations combined with complementary data such as seismic velocities or conductivity models are necessary if a finer resolved mantle structure shall be calculated. The incorporation of such data in the assimilation will also further reduce the uncertainty of the estimated parameters. Furthermore, observations that are independent of the ice history help to reduce errors due to uncertain ice models.

*Data availability.* The synthetic data set used in this study has been submitted to a data repository at Helmholtz Centre Potsdam GFZ. The DOI will be made available when the paper is accepted.

**Appendix A: Basic equations**

In the following we list the basic equations for the response of a self-gravitating Maxwell viscoelastic, incompressible sphere to surface mass load. The complete set of equations and the spectral-finite element approach can be found in Martinec (2000). The following equations are taken directly from that paper.

Consider a viscoelastic sphere $B$ with shear modulus $\mu$ and dynamic viscosity $\nu$. A load represented by the surface density $\sigma$ is placed on $\partial B$. The viscoelastic response of $B$ to the surface mass load is governed by the equation of linear momentum conservation and by Poisson's equation for small perturbations of a hydrostatically pre-stressed and self-gravitating continuum in a non-rotating reference frame:

$$\operatorname{div}\boldsymbol{\tau} - \rho_0\operatorname{grad}\phi_1 + \operatorname{div}(\rho_0\boldsymbol{u})\operatorname{grad}\phi_0 - \operatorname{grad}(\rho_0\boldsymbol{u}\cdot\operatorname{grad}\phi_0) = 0 \quad \text{in } B, \tag{A1}$$

$$\nabla^2\phi_1 + 4\pi G\operatorname{div}(\rho_0\boldsymbol{u}) = 0 \quad \text{in } B, \tag{A2}$$

where $\boldsymbol{\tau}$ is the Cauchy-stress tensor $\boldsymbol{u}$ is the displacement vector, $\phi_1$ is the sum of the perturbation of the initial gravitational potential $\phi_0$ and the potential of the externally applied gravitational force field, $G$ is Newton's gravitational constant, and $\rho_0$ is the unperturbed mass density.

The fundamental properties of $B$ are those of an incompressible Maxwell viscoelastic material:

$$\dot{\boldsymbol{\tau}} = \dot{\boldsymbol{\tau}}^E - \frac{\mu}{\nu}(\boldsymbol{\tau} - \Pi\mathbf{I}) \quad \text{in } B, \tag{A3}$$

$$\boldsymbol{\tau}^E = \Pi\mathbf{I} + 2\mu\epsilon \quad \text{in } B, \tag{A4}$$

where $\epsilon$ is the symmetric part of $\operatorname{grad}\boldsymbol{u}$, $\Pi$ is the perturbation pressure, $\mathbf{I}$ is the second-order identity tensor and the dots above symbols denote time derivatives. The incompressibility of the medium is expressed via the constraint

$$\operatorname{div}\boldsymbol{u} = 0 \quad \text{in } B. \tag{A5}$$

At an internal discontinuity $\Sigma$ the interface conditions for displacement, traction, the perturbed gravitational potential and the perturbed gravitational intensity are

$$[\boldsymbol{u}]_-^+ = 0 \quad \text{and} \quad [\boldsymbol{n}\cdot\boldsymbol{\tau}]_-^+ = 0 \quad \text{on } \Sigma, \tag{A6}$$

$$[\phi_1]_-^+ = 0 \quad \text{and} \quad [(\operatorname{grad}\phi_1 + 4\pi G\rho_0\boldsymbol{u})\cdot\boldsymbol{n}]_-^+ = 0 \quad \text{on } \Sigma, \tag{A7}$$

where $\boldsymbol{n}$ is the outward unit normal on $\Sigma$, the symbol $[f]_-^+$ indicates the jump of quantity $f$ on $\Sigma$ and a superscript denotes the evaluation of $f$ on the exterior $(+)$ of interior $(-)$ side of $\Sigma$.

Initial and boundary conditions are prescribed on the external surface of $\partial B$:

$$\boldsymbol{e}_r\cdot\boldsymbol{\tau}^-\cdot\boldsymbol{e}_r = -g_0(a)\sigma \quad \text{on } \partial B, \tag{A8}$$

$$\boldsymbol{\tau}^-\cdot\boldsymbol{e}_r - (\boldsymbol{e}_r\cdot\boldsymbol{\tau}^-\cdot\boldsymbol{e}_r)\boldsymbol{e}_r = 0 \quad \text{on } \partial B, \tag{A9}$$

$$[\phi_1]_-^+ = 0 \quad \text{on } \partial B, \tag{A10}$$

$$[\operatorname{grad}\phi_1]_-^+\cdot\boldsymbol{e}_r + 4\pi G\rho_0^-(\boldsymbol{u}\cdot\boldsymbol{e}_r) = 4\pi G\sigma \quad \text{on } \partial B \tag{A11}$$

where $\boldsymbol{\tau}^-$, $\rho^-$, and $\boldsymbol{u}^-$ denote stress tensor, unperturbed density, and displacement on the interior side of $\partial B$, respectively, $\boldsymbol{e}_r$ is the unit vector in radial direction, $a$ is the radius of sphere $\partial B$, and $g_0(r)$ is the initial gravitational acceleration at radius $r$.

To summarise: "...the initial boundary value problem for the determination of the displacement field $\boldsymbol{u}$, the perturbed gravitational potential $\phi_1$ and the perturbed pressure $\Pi$ within the viscoelastic sphere $B$ is governed by the partial differential equations (A1) and (A2) constrained by eqs (A3)–(A5), which are valid in $B$ at time $t \geq 0$, by the interface conditions (A6) and (A7), which are applied at the internal interfaces $\Sigma$, and by the boundary conditions (A8)–(A11), which are applied at the external surface $\partial B$ at time $t \geq 0+$" (Martinec, 2000).

### Appendix B: Basic sea level equations

In the following we list the fundamental equations governing sea level behaviour. The equations are taken from Kendall et al. (2005).

Sea level $SL$ is defined globally and the difference between the radial position of the geoid, $G$, and the solid Earth surface, $R$:

$$SL(\theta, \psi, t_j) = G(\theta, \psi, t_j) - R(\theta, \psi, t_j), \tag{B1}$$

where $\theta$ is colatitude and $\psi$ is longitude. Topography is defined as the opposite of globally defined sea level:

$$T(\theta, \psi, t_j) = -SL(\theta, \psi, t_j). \tag{B2}$$

The ocean depth is the projection of the global sea level on to the ice-free oceans. It can be written as

$$S(\theta, \psi, t_j) = SL(\theta, \psi, t_j) \cdot C(\theta, \psi, t_j)\beta(\theta, \psi, t_j), \tag{B3}$$

where the ocean function $C$ is defined by

$$C(\theta, \psi, t_j) = \begin{cases} 1 & \text{if } SL(\theta, \psi, t_j) > 0 \\ 0 & \text{if } SL(\theta, \psi, t_j) \leq 0, \end{cases} \tag{B4}$$

and the $\beta$-field, prescribed from the input ice model, is given by

$$\beta(\theta, \psi, t_j) = \begin{cases} 1 & \text{where there is no grounded ice} \\ 0 & \text{where there is grounded ice.} \end{cases} \tag{B5}$$

GIA-induced perturbations to the geoid and solid surfaces, denoted by $\Delta G$ and $\Delta R$, respectively, lead to variations in sea level. After the onset of loading at time $t_0$, the geoid and the solid surface are given by the sum of the equilibrium geoid / solid surface and the GIA-induced changes to the geoid / solid surface

$$G(\theta, \psi, t_j) = G(\theta, \psi, t_0) + \Delta G(\theta, \psi, t_j) \tag{B6}$$

$$R(\theta, \psi, t_j) = R(\theta, \psi, t_0) + \Delta R(\theta, \psi, t_j). \tag{B7}$$

It follows that

$$SL(\theta,\psi,t_j) = SL(\theta,\psi,t_0) + \Delta SL(\theta,\psi,t_j) \tag{B8}$$

where

$$\Delta SL(\theta,\psi,t_j) = \Delta G(\theta,\psi,t_j) - \Delta R(\theta,\psi,t_j) \tag{B9}$$

Using (B2), (B3), and (B8) one can derive the generalised sea level equation

$$\Delta S(\theta,\psi,t_j) = \Delta SL(\theta,\psi,t_j)C(\theta,\psi,t_j) - T(\theta,\psi,t_0)[C(\theta,\psi,t_j)\beta(\theta,\psi,t_j) - C(\theta,\psi,t_0)\beta(\theta,\psi,t_0)]. \tag{B10}$$

*Author contributions.* RS has been responsible for the methodology, investigation, formal analysis, visualization, and preparation of the
605 initial manuscript draft. JS was responsible for conceptualization, methodology, project administration, and project supervision.VK has
contributed the code for the VILMA model and supported the methodology. MB has provided auxiliary software, and supported the investi-
gation. MT has been involved in project administration and funding acquisition. All authors have been involved in reviewing and editing the
manuscript.

*Competing interests.* The authors declare that they have no conflict of interests.

*Acknowledgements.* The work described in this paper has received funding from the Initiative and Networking Fund of the Helmholtz
Association through the project "Advanced Earth System Modelling Capacity (ESM)". The numerical simulations were performed at the
German Climate Computing Center (DKRZ). Parts of the study are based on work funded by the German climate modeling project PalMod
(FKZ: 01LP1502E, 01LP1503A) and PalMod II (FKZ: 01LP1918A) that was supported by the German Federal Ministry of Education and
Research (BMBF) as a Research for Sustainability initiative (FONA). Some figures were produced using the Generic Mapping Tool (GMT)
by Wessel et al. (2019).

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
