# Peer review of "An approach for constraining mantle viscosities through assimilation of paleo sea level data into a glacial isostatic adjustment model"

_Nonlinear Processes in Geophysics, 2021_

## Author Comment (AC1)

Dear reviewer,

we are thankful for the constructive criticism and the suggestions made about the manuscript. In the following you find detailed answers to the points that were raised.

Reviewer's comments and author's answers:

"An approach for constraining mantle viscosities through assimilation of paleo sea level data into a glacial isostatic adjustment model" by Schachtschneider et al., present a data assimilation approach to recover mantle viscosity parameters for a 3-layer glacial isostatic adjustment (GIA) model using synthetic relative sea-level change data. Based on a range of experiments, they conclude their particle filter based method was able to find the ground truth mantle viscosity parameters given different initial ensemble probabilities, observational uncertainties and spatial-temporal data coverage. This is a novel application of the data assimilation approach for GIA modelling studies, which has a good potential to help the community to better constrain the mantle viscosity parameters of a commonly-used 3-layer GIA model. This is a well-written manuscript which explains a complex subject in a very clear way. I have two major points and other minor comments. Provided the authors address these points in a subsequent revision I would certainly support the publication of this manuscript.

Major points

Firstly, given this is mainly a methodological paper for finding the mantle viscosity parameters, it is unclear what the major advantages of this data assimilation approach are compared to other commonly-used statistical approaches for finding the best-fit GIA model parameters. For example, a common approach is to use chi-square misfits to find the confidence interval of mantle viscosity (see Lambeck et al., 2014), which allows the best-fit GIA parameters to locate anywhere within the initial distribution. The authors have mentioned that their approach can converge to viscosity values that are not covered by the initial ensemble, which can also be approximately achieved given a denser sampling of possible viscosity parameter values within a forward modelling scenario. In this case, the computation time for this new approach is a very important information to report. The authors should at least demonstrate one advantage of this new approach either regarding the inversion power or the computational efficiency.

*A: Our approach is slightly slower than a pure model run with the same ensemble size. This is due to the overhead of the particle filter in which some steps have to be performed in serial mode. However, due to the ability to change the parameters of the ensemble members, smaller ensemble sizes and thus smaller total computing time in terms of node hours are possible while still obtaining very accurate parameter estimates. We have now emphasized the advantages of our method more strongly in the manuscript.*

Secondly, the experiment set-up is over-idealized. In order to apply this method for real GIA problems in the future, I suggest the authors consider relaxing several assumptions, or at least discuss how relaxing these assumptions will impact the final results:

1. The authors assume a temporally-uniform relative sea level (RSL) data uncertainty of

0.5/0.25/0.1 m throughout the Last Deglaciation, which is however very difficult to achieve in reality. The RSL data uncertainty tends to be larger for earlier time intervals (e.g., a large proportion of pre-Holocene RSL data are coral-based records, which usually have >2 m uncertainty range) and gradually decreases for more recent time intervals as more stratigraphy-based data became available. Therefore, the assumption of 0.5 m upper limit for RSL data is not solid, and it is important to check how a temporally variable (sea level index point) SLIP uncertainty distribution and a larger SLIP uncertainty would impact the assimilation results.

*A: Valid point. New experiments with larger uncertainties that decrease towards present day have been conducted (new case E of setup 1). The size of observation uncertainties were chosen according to values given in Vacchi et al. (2018) and Khan et al. (2019). Our chosen values are even more conservative than the values given in the appendix of Lambeck et al. (2014). The obtained results were added to the manuscript. The viscosity values still converge to the given ground truth (target values) with a larger ensemble spread according to the assumed observation uncertainties. In order to investigate other influences on the ensemble convergence we use a more idealistic setup such that other effects are not masked by observation uncertainty.*

2. The authors assume a $\sigma_{init}$ of $2\times10^{20}/2\times10^{19}$ Pa s for lower/upper mantle viscosity, which is a very small range given that the authors suggest the initial ensemble should cover the ground truth parameter value (c.f. the Lambeck et al., 2014 search space is $10^{19} - 10^{21}$ and $5\times10^{20} - 10^{24}$ Pa s for upper and lower mantle respectively). My query is what if the authors use a larger $\sigma_{init}$ value, would it affect the final assimilation results?

*A: This experiment uses synthetic data from an artificial ground truth. It is only an example and does not need to be equal to reality to show the success of the method. We aim at showing that we can recover the target values of a viscosity model on which the synthetic observations are based. Our assumed ground truth differs from values currently assumed valid by the community but that does not affect the results of the study.*
*Using a larger spread in the initial ensemble is possible and leads to the same assimilation results. However, one must be careful not to end up with a degenerating filter. That can happen if the spread is loo large and only few (or even only one) ensemble member is close enough to the true values to have a significant likelihood. Larger values for observation uncertainty in the beginning of the assimilation (as shown in the new case E of setup 1) or a larger ensemble with denser sampling of the model parameter distribution help to prevent filter degeneracy since in that case models that are quite far from the ground truth can still be have significant likelihood values. If observation uncertainties decrease towards present day, the ensemble spread decreases too and parameter estimations with low uncertainties can be obtained. This was added to the discussion in the manuscript.*

3. The authors assume no ice loading history uncertainty in the synthetic experiments, but the ice loading history is the biggest uncertainty for GIA modelling problems. It would be useful to discuss the potential problems caused by uncertain ice loading history, in other words, are there some possible solutions for the situation if we are uncertain about where to propagate the particles?

*A: This is indeed a major source of uncertainty when the approach will be applied to real*

*sea-level data. A more detailed discussion of this point was added to the manuscript. A comprehensive model set up is beyond the presented study, but the consideration of load history uncertainties is planned for a follow up study.*

Minor points:

-l25: The authors should pay attention to the differing role of mantle viscosity in controlling RSL change in the near-field and far-field regions. For example, in line 25 here, on-going GIA-governed far-field sea level change is primarily due to the 'fingerprint' effect (i.e., the spatially variable RSL change associated with changes to the shape of the gravitational field), which is largely not related to the mantle viscosity parameters. The mantle viscosity parameters are more important for near-field regions where viscoelastic land deformation dominate the local RSL change. Similarly, for line 186, the Earth deformation only dominates the near-field region, not RSL change for all regions. I would suggest the authors including some statements about the RSL change differences between the near- and far-field regions and highlight that the mantle viscosity parameter are more important for the near-field RSL change.

*A: Statements regarding the difference of the meaning of RSL change in the near and far field were added.*

-l65: What is the point of this paragraph? do you use this approach or not?

*A: The goal was to explain our motivation for using the Particle Filter. I agree that this does not become clear and this paragraph was combined with the next one and rewritten.*

-l69: Including a reference for the particle filter here would be useful.

*A: A reference to section 3.2 where particle filters are described was added.*

-l93: What does the subscript i mean for equation 2?

*A: The subscript indicates the particle member in the ensemble. An explanation was added.*

-l130: How does the ensemble propagate through time? Do the authors run 50 forward GIA models with different mantle viscosity parameters or just one reference model to guide the propagation length and direction? I think you will update the mantle viscosity parameters during the deglaciation experiment. In order to calculate the RMS for that ensemble member, do you go back and calculate RSL value for this updated model from the start of the experiment (25.5/9.5 ka BP)?

*A: The ensemble consists of 50 independent members with different viscosity values. Each member is propagated through time individually. When observations are available, the fitness of each member is estimated based on the difference between model prediction and observations. Members with poor performances are discarded in favor of model states of better performing members. The latter are cloned and perturbed to reestablish an ensemble of 50 members. Then, the propagation through time is continued. The models are not restarted from t_0. A reference to section 3.2, where the ensemble update is described, was added.*

-l146: It would be useful to mention that RSL rate is a non-standard type of observation; it is more common to compare model output with absolute RSL observations. When dealing with real-world data, the uncertainty on RSL rate will be greater than the uncertainty on absolute RSL due to the cumulative impacts of spatial-temporal uncertainty on the individual observations.

*A: The fact that RSL rates are non-standard was mentioned. The increased uncertainty of RSL rates was already considered in our experiments. It is now also mentioned in the manuscript. As stated in the manuscript, we have used sea-level rates instead of relative sea level due to the fact, that we do not know the sea level at present day during our integration scheme.*

-l167: You quote the uncertainty on RSL observations, but you compare model output with RSL rates. It would be useful to know the RSL rate uncertainty (m/ka) produced by 0.1 and 0.5 m RSL uncertainty.

*A: Indeed, this is a vital piece of information. There is a factor \sqrt{2} / \delta t between RSL  uncertainties and rate uncertainties. The information was added to the manuscript.*

-l169: Please define what you mean by 'initial offset' here.

*A: The term "initial offset" refers to initial perturbation of the ensemble members at the beginning of the assimilation. It consists of a random value drawn from a  normal distribution with mean \mu_{init} and variance \sigma^2_{init}. With "offset" we describe the mean of  the initial perturbation. (see l. 125 in the original manuscript). The function of \mu_{init} and \sigma_{init} are now mentioned in the first paragraph of Sect. 4.1 and a reference to that place was added here.*

-l198: Since the synthetic observations are RSL rate, why are the RMS values expressed in m instead of m/ka?

*A: This is indeed a typo. The labels were changed to the correct unit "m/kyr".*

-Figure5: Can the authors slightly expand on why there are RMS spikes after meltwater pulses? Specifically, meltwater pulses are produced by large ice mass loss from North American and Fennoscandia Ice Sheets which will produce a large signal of land deformation in those regions (which could be larger than the observational uncertainty), this will help the assimilation process to find the optimal solution. Therefore, given a better converged mantle viscosity ensemble after meltwater pulses, why are there such large spikes immediately after the improvement in the mantle viscosity solution?

*A: During phases of strong melting like melt-water pulses, the sea level rises instantaneously during one assimilation step, resulting in a significant global sea-level change. This change is not caused or affected by any relaxation process. Accordingly, all viscosity structures would reproduce this change, given the ice melt is predefined in our assimilation experiment. In the following assimilation steps, the sea level rate is again governed by the relaxation process of the solid earth, which forces the enlarged ensemble spread to converge again to the best fitting subset. A more detailed explanation was added in the discussion.*

-l225: Both the selected regions for Setup 2 use near-field region SLIPs where RSL

change are sensitive to the mantle viscosity value. If the authors just used the far-field region SLIPs, does the assimilation approach work as well? If not, the authors should note their method should only be applied to near-field GIA problems.

*A: Tests with only far-field observations were not successful. This is now mentioned in the text .*

-l259: "As a consequence less models…", should use "fewer models" here.

*A: Corrected. Thank you!*

-l296: "There are no large ice mass changes after 10 ka BP." However, based on Figure 7 of this paper, the continental ice volume was still changing after between 10 and 5 ka BP.

*A: The sentence was rephrased to express that ice mass change rate was already low that 10 ka BP and Fennoscandian and Laurentide ice shields are basically vanished by 8 ka BP. After that, ice loss can still be seen in Greenland and Antarctica.*

-l318: It would be useful if more details about computation time can be given here.

*A: Computing time strongly depends on the used computer system. Therefore, comparisons can only be made to methods tested on the same system which was not part of this study. However, we can state that due to the overhead of the particle filter, running time is larger than for the dynamic model alone. The amount of overhead depends on the ensemble size since some parts of the assimilation are performed sequentially. In our case with an ensemble size of N=50 the overhead is about 30–40 % depending on the physical allocation of compute nodes in the cluster. This estimation was added in the text.*

-l324: How would the temporal variability of the data availability affect the results?

*A: We added a scenario in which only observations from ice-free locations were used. This reduces the available observations drastically for early periods. As a result, the convergence is very slow in the beginning of the assimilation period but increases after about 10 kyrs BP. After that we observe good convergence and are able to recover the target viscosity values.*

-l325: "… ~8 to 12 ka BP and is considerable smaller", should be considerably smaller here.

*A: It is corrected, thank you.*

-l337: I suggest the authors cite Khan et al., (2019) as the reference for the HOLSEA community here.

*A: The reference was added.*

---

## Author Comment (AC2)

Dear Peter Jan van Leeuwen,

we are thankful for the constructive criticism and the suggestions made about the manuscript. In the following you find detailed answers to the points that were raised.

Reviewer's comments and author's answers:

The manuscript discusses the estimation of mantel viscocities using sea-level observation in a particle filter. The manuscript is well written, and the results are interesting even though the set-up is highly simplified. I suggest publication after the following minor comments are taken into account.

Line 78: Strange sentence, not sure what the authors want to convey.

*A: The aim of the sentence was to explain the name of particle filter. It was removed.*

Line 79: Note that the output of the filter is the weighted posterior ensemble, from which a weighted mean can be calculated. This mean can be a poor estimate of the posterior pdf if that pdf is strongly non-Gaussian. Please add a small discussion of these facts.

*A: The description of the filter output has been changed in the manuscript. A short discussion about the posterior pdf estimate and the situation in this study was added.*

Line 102: Resampling hardly changes the ensemble variance of the weighted ensemble, which is the relevant ensemble in this case. The weighting itself reduces the variance in the ensemble.

*A: The sentence was changed and now states that weighting reduces the ensemble variance.*

Line 105: It would be good to mention that of the three methods the second is the correct methods, and the 1st and 3rd are approximations.

*A: The fact was added to the text.*

Line 111: I assume the authors mean $N(0, a\sigma^U, L)$. Note that it is common to have the (co)variance of the distribution as the second argument of $N(..,..)$ and not the standard deviation.

*A: It was changed to standard notation here and in subsequent occurrences in the text.*

Line 114: Setting $a = 0.5\sigma$ is a large value to perturb each viscosity with. However, it is perhaps good to mention that the standard deviation of the ensemble as a whole only increases by a factor 1.12 (= sqrt(1 + 0.25)) through this procedure.

*A: The mentioned piece of information was added in terms of variance (increase of variance by 25%) for consistency.*

Line 115: Probabilistic resampling means that one has to draw N random numbers, where

N is the ensemble size. A more accurate resampling method is Stochastic Universal Resampling, in which only one random number is drawn, and it is also faster! This could be mentioned. Since after resampling relatively large random perturbations are added to the particles the difference will be minor in this case.

*A: Stochastic Universal Resampling as another possibility was mentioned in the text and a discussing sentence has been added..*

Line 165: mu
*A: The typo was corrected.*

Line 174-175: Please remove this sentence, it is just a repetition of what was said before.

*A: Agreed. The sentence was deleted.*

Since the variance in the initial ensemble is chosen as large as the mean many initial particles will have negative viscosities. What is done when a negative viscosity is drawn? A similar question for later in the run, what is done if the 'jittering' after resampling produces negative viscosities?

*A: In those cases mentioned, the absolute value was used in case of a resulting negative viscosity.*

Related to this, the figures show different mean viscocities than the table 2. Please correct the one that is incorrect.

*A: The table caption was not accurate. Shown in Table 2 are mean and standard deviations of the initial perturbations that were added on top of the target viscosities. The table caption was corrected.*

It would be good to show the effective ensemble size, defined as $N\_eff = 1/sum\_i (w\_i^2)$ in which the $w\_i$ are the normalized weights, such that $sum\_i w\_i = 1$. This allows the readers to judge the quality of the ensemble. My suspicion is that $N\_eff$ is rather low, as low as 2-5 members at time, which is close to degeneracy.

*A: A figure showing the effective ensemble size was added. The effective ensemble size is mostly quite high. In the presented case (cases A, B, and C in setup 1), it only drops right after a big melt water pulse when also the residuals of the predictions w.r.t. observations are very high. In the newly added case E in setup 1, it drops significantly when the observation uncertainties are reduced since in that case suddenly a large number of ensemble members have bad performance values (low likelihood). Nevertheless, the effective ensemble size increases in the next assimilation step.*

Fig 10: caption, change left and right to top and bottom.

*A: Done. Thank you!*

Line 309: I don't understand this sentence, please clarify.

*A: "Common approach" in viscosity determination from GIA means that an ensemble of, say, 50 members with pre-defined viscosity structures is used in a run of forward models. Paleo-sea level observations are then used to determine the best fitting model via RMS*

*errors of sea level predictions. This way, only the best model of the initial ensemble can be determined. With our approach we can obtain a model that was not in the initial ensemble and fits the observations best. The sentence was rephrased a bit to make it clearer.*

Section 2: I'm not an expert in this field and would suggest providing the set of equations being solved to gain an idea of the complexity of the problem at hand.

*A: The complete set of equations that are solved can be found in the references given in the respective parts of the manuscript. However, we have added the most basic equations in the appendix.*

Section 7: It would be good to also include a discussion of the accuracy of the underlying ice model and its expected influence on the results.

*A: Naturally, the forcing due to the ice load plays a key role in GIA. Therefore, uncertainties of the ice model have a large impact on the resulting viscosity values. However, uncertainties for ice models are usually not provided. We added some discussion about the possible influence of ice load errors and the resulting magnitude of viscosity uncertainty but a comprehensive analysis was not part of this study. Ice model uncertainty is also a problem in conventional viscosity determination and does not distinguish our approach from the conventional one. Therefore, we focused on the methodology of our approach and refrain from a detailed analysis of this matter.*

---

## Author Comment (AC3)

Dear Dan Crisan,

we are thankful for the constructive criticism and the suggestions made about the manuscript. In the following you find detailed answers to the points that were raised.

Reviewer's comments and author's answers:

The authors use a particle filter for parameter estimation in a visco-elastic model of the lithosphere and mantle. The goal here is to find a set of parameters in a model that leads to a solution consistent with a set of observations. In this work the unknown parameters are the viscosities of the lower and upper mantle. The authors study the effect of different parameter initializations and observation uncertainty on the performance of the filter with clearly stated results and conclusions.

The paper is well organized and well-presented. I recommend publication but suggest that the authors address the following niggles:

Line 78 I don't get this sentence "the ensemble members did not mix they were called particles"

*A: The intention of this sentence was to explain the name particle filter. Since it was also unclear to other reviewersit was removed.*

Line 85 This is also a very good review paper:

PJ Van Leeuwen, HR Künsch, L Nerger, R Potthast, S Reich, Particle filters for high-dimensional geoscience applications: A review, Quarterly Journal of the Royal Meteorological Society 145 (723), 2335-2365, 2019

*A: Actually, we had also consulted this paper. It was added here as a reference.*

To find more generic references on particle filters, I suggest that the authors look at:

Doucet, A.; Johansen, A. M. A tutorial on particle filtering and smoothing: fifteen years later. The Oxford handbook of nonlinear filtering, 656–704, Oxford Univ. Press, Oxford, 2011.

*A: Thank you for this hint.*

Line 110 An alternative method to update the parameter set is to use jittering. For details:

Crisan, Dan; Míguez, Joaquín Nested particle filters for online parameter estimation in discrete-time state-space Markov models. Bernoulli 24 (2018), no. 4A, 3039–3086.

*A: The jittering method has been added in the manuscript as another possibility to update the parameters.*

Line 138 In section 4.2 It would be good to explain what the observations are, the choice of the distribution for observation noise (the observation uncertainty parameter is stated at line 165 but not the distribution) , and give the  expression for the parameter likelihood function.

Finally, perhaps you can put the VILMA equations in an appendix.

*A: The complete set of equations that are solved can be found in the references given in the respective parts of the manuscript. However, we have added the most basic equations in the appendix.*

---

## Author Comment (AC5)

Dear reviewer,

we are thankful for the constructive criticism and the suggestions made about the manuscript. In the following, we give detailed answers about the raised questions.

Reviewer's comments and author's answers:

This manuscript introduces a combination of a data assimilation method with a GIA model. The new combination approach is supposed to, ideally, better determine mantle viscosities or, in future, 3D earth parameters. To achieve this, a set of global synthetical relative sea level rates is generated. The authors perform a number of tests to convince the reader about the favourable outcome of their new approach. They seem to be new in the field of GIA modelling. I am not aware of any previous GIA work with the exception of Volker Klemann, who has a strong background in this field.
The study is very interesting and the approach may receive much interest in the GIA and earth rheology communities. The text is well written and a smooth read. Figures and tables are clear and support the text.

Nonetheless, I am somewhat disappointed about the whole manuscript. Main reasons are that, despite the nice presentation and different tests made by the authors, (1) nothing is presented about the performance of the new approach compared to 'GIA standard' methods, (2) no interesting conclusions are drawn that present a step forward in GIA/earth rheology research, and (3) it is a very idealized experiment because the synthetic data does not represent typical data used in GIA modelling. RSL rates is not used, moreover, rates are hard to find in real data as there is rarely a large number of samples at a certain location available. Also, uncertainties of rates are much larger than used in these setups. Overall, the main message is that the technical combination of two codes is working and gives results that are expected. The results thus, at this stage, cannot help to further advance our understanding of GIA or earth parameters.

However, I think the manuscript can be elevated if my main concerns can be addressed.

(1) Especially, I would like to see a comparison with a 'standard' GIA investigation, where modelled RSL rates from a set of pre-defined models (e.g., 50 models covering the viscosity ranges in your experiment) is compared to the synthetic set and the misfit is determined so that a best model of such set is identified. Is the best-fitting model comparable to the final assimilation model? Which misfit is better? What is the computation time for both approaches? At which point is it better to use the assimilation approach? This would help the reader to get more perspective if this approach can help advance our understanding of GIA and the determination of earth parameters.

*A: While we have to admit that this is a really interesting piece of information, it is hard to make this comparison with our study. We have used synthetic observations from a synthetic scenario. Making comparison to a real-world model is hardly possible. We can, however, give RMS errors of the modeled RSL which can then be compared to RMS*

*errors from literature. One has to keep in mind that the resulting RMS errors depend on the assumed observations uncertainties. In the update of our manuscript we have added an experiment with more realistic RSL uncertainties (case E in setup one) which give a better estimate of the accuracy that can be expected when using the DA approach with real data. The sea-level RMS errors from this experiment E are now used for a comparison to viscosity estimations using real observations.*

(2) I miss new findings or hints that can help the community. The manuscript presents the approach with some tests, which gives it the style of a technical note rather than a scientific study. The authors should at least present 1 or 2 major conclusions that can be drawn from the tests.

*A: The aim of this paper is to show that the data assimilation approach is a versatile method that is able to estimate the correct mantle viscosities from a synthetic Earth model. We have shown that we obtain the correct mantle parameters within an uncertainty range defined by the quality of the observations. With our method we can obtain model parameters that are not part of the initial guess, but the ensemble members can evolve towards the correct solution. This is very different from the classic approach. Especially, when going towards higher-dimensional parameter spaces, e.g., higher resolved 1D profiles or 3D viscosity distributions, this will be very helpful.*

(3) The reliability of the rates set should be further discussed in comparison to real world data. You mention some shortcomings but they are not put into perspective with real data availability. How many locations can actually give you solid RSL rates? What is a realistic error of such RSL rates? This should definitely be addressed as RSL data are concerned with time and height errors. You did not include time errors which are actually much larger! Are there enough locations with rates at times where there is a strong RSL fall? Such discussion would help the reader to get more insight on the reliability and evaluate the success of your approach.

*A: From the number of available sites which are commonly used to reconstruct the temporal evolution of the sea level from the late Pleistocene or Holocene to present day, about 20 to 30 % contain more than 10 samples. This percentage might suite as an estimation of the availability of sites which can be used to derive past rates of sea-level change. On the other hand e.g. the study of Khan et al. (2015) lists average RSL rates for a large number of locations, indicating that rates are available at least for the last 10 to 12 kyrs. This is now mentioned in the manuscript.*

(4) A discussion is needed on the tested parameters. Just analyzing two mantle viscosities is very idealized. There is a trade-off between the thickness of the lithosphere and mantle viscosity. The reader should be informed.

*A: A short discussion of the trade-off between lithosphere thickness and mantle viscosity has been added. Unfortunately, at the moment our approach does not allow to vary lithosphere thickness. Therefore, we focused on mantle viscosity and kept the lithosphere thickness constant.*

Similarly, a note on ice model uncertainty and its potential impact on the results should be added.

*A: Uncertainty in GIA is a big problem. Usually, no uncertainties are provided for global ice models. Ice histories from different approaches (e.g., ICE-5G by Peltier (2004), ICE-6G by Argus et al. (2015), PaleoMIST 1.0 by Gowan et al. (2021), and NAICE by Gowan et al. (2016)) reveal large deviations between ice distributions (thickness and extension) during deglaciation. A different ice load significantly affects the outcome of the viscosity determination. A short discussion of these relations was added.*

Minor remarks

The paper is written from a quite technical perspective. In the introduction, focus is a lot on the assimilation approach but I would like to see a paragraph from the 'GIA site' with an overview of previous attempts to get more insights from GIA modelling with alternate approaches. Studies by Steffen & Kaufmann (2005), Al-Attar & Tromp (2013) and Caron et al. (2017) should help here.

*A: A paragraph describing efforts to determine mantle viscosity through GIA modelling or sea level observations was added in the introduction.*

Similarly, the discussion does not contain much references to other works. Are all these findings/conclusions new?

*A: While data assimilation is used in GIA to estimate past sea levels, we are not aware of any other study attempting to constrain mantle viscosity with a particle filter. References to other attempts to infer mantle viscosities by means of data assimilation in general have been added to the introduction.*
*/*

References

Al-Attar, D., Tromp, J., 2013. Sensitivity kernels for viscoelastic loading based on adjoint methods. GJI, doi:10.1093/gji/ggt395.

Caron, L. et al., 2017. Inverting Glacial Isostatic Adjustment signal using Bayesian framework and two linearly relaxing rheologies. GJI, doi: 10.1093/gji/ggx083.

Steffen H., Kaufmann, G., 2005. Glacial isostatic adjustment of Scandinavia and northwestern Europe and the radial viscosity structure of the Earth's mantle. GJI, doi:10.1111/j.1365-246X.2005.02740.x.

---

## Editor Decision (ED1)

Dear Colleague,

Two referees have submitted their reviews of the revised version of your paper. They are referees 3 and 4 of the first version, with the same identification numbers.

In particular, referee 3, who has again let his name known, is Dan Crisan. He recommends acceptance of the paper as it stands.

Referee 4 asks for minor revisions. He/she considers (main comment) that the comparison he/she had asked for in the first place should not be difficult to make. He/she also suggests a number of editing corrections.

Please consider carefully the referee's main comment. Contrary to what you write (*Making comparison to a real-world model is hardly possible*), he/she is not asking you to use a model different from the one you have used. He/she is only asking you to run your model 50 times, with varying values for the viscosity parameters. That should certainly be very instructive, and, from what I understand, should not be costly. If you consider that the referee's request is too costly or difficult, please explain more clearly why you think so.

I also have as editor a number of suggestions for minor corrections (the line numbers, as in referee 4's comments, are the ones of the Author's tracked changes (ATC) version of the paper).

- L. 65, *we facilitate*, do you mean *use* ?
- Ll. 69-70, sentence beginning *The sea level* …. is difficult to understand (syntax is clumsy).
- Caption of Fig. 6 (and other figures that follow) *Development of viscosity* …. I think *Variations of viscosity in the course of the assimilation* … would be preferable (see also l. 261)
- L. 128, I presume there must be a minus sign in the argument of the exponential.
- L. 129, *observation residuals* have not been defined at this stage (see ll. 135-136)
- L. 150, *We follow the first approach*. You have at this stage defined four approaches, including *jittering*. Clarify.
- L. 160, formula defining $w'_i$. The weights $w_j$ are to be ranked in increasing order.
- L. 162, *Stochastic Universal Resampling*. Explain what that is (or at least give reference)
- L. 229, consistency of notation would require $N(\mu_{init}, \sigma^2_{init})$ (as on l. 172)
- L. 297, *The variance or STD* …You show only standard deviation (*STD*) in what follows. Don't mention variance at this stage.

I had a quick look at the Appendices, and some editing is also to be done there

- The quantity $\phi_0$ in Eq. (A1) is apparently not defined
- How is the perturbation pressure $\Pi$ (Eqs A3-4) known ?
- L. 584, *inverse* → *opposite*,
- I suggest you check the equations in Appendix B. Although they must be elementary, they seem erroneous in some respects (I am intrigued for instance by the product $DG(,,) R(,,)$ in Eq. B6).

Please revise your paper taking into account all comments and suggestions of referee 4, as well as mine. In case you disagree with a particular comment or decide not to follow a particular suggestion, explain your reasons.

I look forward to receiving the revised version of your paper. I may submit it to referee 4 for further review.

---

## Author Response (AR2)

Dear Olivier Talagrand,

we have considered the reviewers request of a comparison with the standard GIA investigation and conducted the respective experiments. The manuscript was updated with the results from those experiments and with the editorial changes suggested by the reviewer 4 and by you with one exception:

Your comment "L. 160, formula defining w' i . The weights w j are to be ranked in increasing order." → In our approach, the order of the weights does not matter and we do not rank them in increasing order (cf. Fig. 2, the normalised weight w_j is the difference between the y-values of particles j and j-1).

Regarding your second remark about the appendix-equations: the perturbation pressure is not known but determined in a boundary value problem. The explaining paragraph in Martinec (2000) reads:

"In summary, the initial boundary value problem for the determination of the displacement field **u**, the perturbed gravitational potential \phi_1 and the perturbed pressure \Pi within the viscoelastic sphere *B* is governed by the partial differential equations (1) and (2) constrained by eqs (3)--(5), which are valid in *B* at time *t* >= 0, by the interface conditions (6) and (7), which are applied at the internal interfaces \Sigma, and by the boundary conditions (8)--(11), which are applied at the external surface \partial *B* at time *t* >=0+." A citation of this text passage was added for clarification.

The "strange" equation B6 was due to a formatting error, it should be two separate equations. It was corrected but is not marked by the latexdiff script. Please have a look.

We thank you and the reviewers for their efforts and hope that the new manuscripts meets the standards and expectations and is ready for publication in NPG.

With kind regards

The authors

Dear reviewer,

thank you for clarifying your request regarding the comparison with a standard GIA investigation. We have conducted the requested experiments and added a subsection in the discussion (Sect. 6.7).
All your editorial suggestions have been considered and the manuscript was changed accordingly.

With kind regards

The authors

---

## Editor Decision (ED2)

Dear Colleague,

I have submitted the latest version of your paper to the referee who had asked additional results (referee #4). He/she is satisfied with this new version, and recommends acceptance of the paper as it stands. I will follow the referee's advice.

I also ask as Editor for two minor changes.

- You still use the word *development* (for instance in the caption of Fig. 5) in places where *variation* would be preferable in my mind.

- The acronym *GRD* (l. 31 of file npg-2021-22-ATC2) does not seem to be expanded in the paper.

I thank you for having submitted your paper to *Nonlinear Processes in Geophysics*.

With regards

---

## Author Response (AR3)

Dear Olivier Talagrand,

thank you for the acceptance of our manuscript in NPG! We have made the editorial changes you requested:

- changed all occurrences of the word "development" to "variation" in the main text and in the figure captions
- added an explanation of the acronym "GRD" in the introduction

We thank you again for your efforts and support in the review process.

With kind regards

The authors